# Molecular features of myosin F adapted for driving actin flows in *Toxoplasma gondii*

**Thomas E. Sladewski\* and Aoife T. Heaslip\***

## ABSTRACT

*Toxoplasma gondii* is a single-celled apicomplexan parasite that relies on a highly polarized endomembrane system for its invasion of, and survival within, host cells. Recent advances in imaging technologies have revealed that vesicle transport and the organization of organelles within the endomembrane pathway require a highly dynamic actin cytoskeleton. These dynamics in turn rely on the activity of myosin F (MyoF), a molecular motor unique to alveolates. The defining characteristic of this molecular motor is a WD40 β-propeller domain, exclusively found in this class of myosin. To understand the mechanism by which MyoF controls the dynamics and organization of actin, we studied the biophysical properties of the purified motor *in vitro*. A MyoF construct lacking its WD40 tail domain (MyoF-Motor) is dimeric and can bind and translocate actin in an *in vitro* motility assay. Single-molecule studies show that the dimeric construct is non-processive; however, small ensembles move inefficiently on single filaments of skeletal actin. In contrast, single molecules of the full-length motor move processively on *Toxoplasma* actin and jasplakinolide-stabilized skeletal actin bundles. Electron microscopy of negatively stained images of MyoF and quantitative size exclusion chromatography show that the WD40 domain oligomerizes to form a complex containing multiple dimeric molecules, which provides an explanation for why the full-length motor is processive compared to the dimeric MyoF-Motor construct. Finally, we show that MyoF binds microtubules through its WD40 domain and can slide actin filaments relative to microtubules. We propose a model whereby MyoF oligomers drive actin dynamics by translocating filaments relative to the cytoskeleton of the parasite. These molecular features provide new insight into how MyoF functions in the cell to regulate actin organization during vesicle transport.

KEY WORDS: Myosin, Actin, *Toxoplasma gondii*, Microtubule

## INTRODUCTION

*Toxoplasma gondii* is an obligate intracellular parasite that infects approximately one-third of the world's population. Infection results in severe disease in immunocompromised individuals and when infection occurs *in utero* (Rorman et al., 2006; Torgerson and Mastroiacovo, 2013). Disease pathogenesis and parasite survival depend on the ability of the parasite to complete multiple rounds of its

Department of Molecular and Cell Biology, University of Connecticut, Storrs, CT 06269, USA.

*Authors for correspondence (aoife.heaslip@uconn.edu; thomas.sladewski@uconn.edu)

T.E.S., 0000-0002-8584-5007; A.T.H., 0000-0002-7838-3617

lytic cycle. This involves host cell invasion, replication, and egress, which ultimately destroys the invaded host cell (Blader et al., 2015). These steps rely on a highly polarized secretion system composed of micronemes and rhoptries, which are localized at the apical end of the parasite, as well as dense granules, which are highly motile vesicles distributed throughout the cytosol (Heaslip et al., 2016). The secreted contents of the three vesicle types allows the parasite to attach and invade host cells (Bisio and Soldati-Favre, 2019; Cova et al., 2022), modulate host immune response pathways (Lima and Lodoen, 2019), establish a chronic infection (Griffith et al., 2022), and permeabilize cell membranes to facilitate egress (Kafsack et al., 2009).

We previously determined that the organization of the endomembrane system in *T. gondii* (prefix *Tg* on protein symbols) is controlled by a divergent actin gene (*Tg*Act1) and myosin F (MyoF), a class of myosin motor found exclusively in alveolate protists, which encompasses the apicomplexan phylum (Mueller et al., 2017). Depletion of either of these proteins results in defects in the movement of dense granules, namely Rab6-, Rab11a- and Rop1-positive vesicles (Heaslip et al., 2016; Venugopal et al., 2020; Carmeille et al., 2021). Other components of the endomembrane system are also affected, including Golgi morphology and positioning, the inheritance of a plastid-like organelle called the apicoplast, and the movement of ER tubules (Heaslip et al., 2016; Carmeille et al., 2021; Devarakonda et al., 2023).

It is our goal to determine the mechanisms by which *Tg*Act1 and MyoF control the positioning of such a wide range of cellular cargoes in *T. gondii*. Live-cell imaging using an actin chromobody revealed that intracellular parasites contain a highly dynamic cytosolic actin network (Periz et al., 2017, 2019; Kellermeier and Heaslip, 2024). Depletion of MyoF results in severely reduced actin dynamics and compaction of the cytosolic actin network into bundled structures apical to the Golgi (Kellermeier and Heaslip, 2024). The requirement of MyoF for filament dynamics indicates that this motor and *Tg*Act1 make up an unconventional acto-myosin system that drives cargo transport in *T. gondii*, which is distinct from the 'canonical' cargo transport described in yeast and mammals whereby molecular motors bind cargo via the C-terminal tail domains and transport it along actin or microtubule cytoskeletal tracks (Vale, 2003; Trybus, 2008; Vale and Fletterick, 1997; Hammer and Sellers, 2011). In this study, we analyze the molecular features and biophysical properties of MyoF to understand how the motor facilitates actin dynamics to drive cargo transport.

## RESULTS

### The speed of MyoF is dependent on light chain composition

MyoF has a well conserved motor domain with the exception of three unique inserts (Fig. 1A,B), the function of which are currently unknown (Heaslip et al., 2016). An AlphaFold model of the MyoF-Motor domain (blue) and lever arm bound to a single calmodulin light chain (gray) shows that the second unique insert in the motor domain is located adjacent to the light chain bound to the first IQ

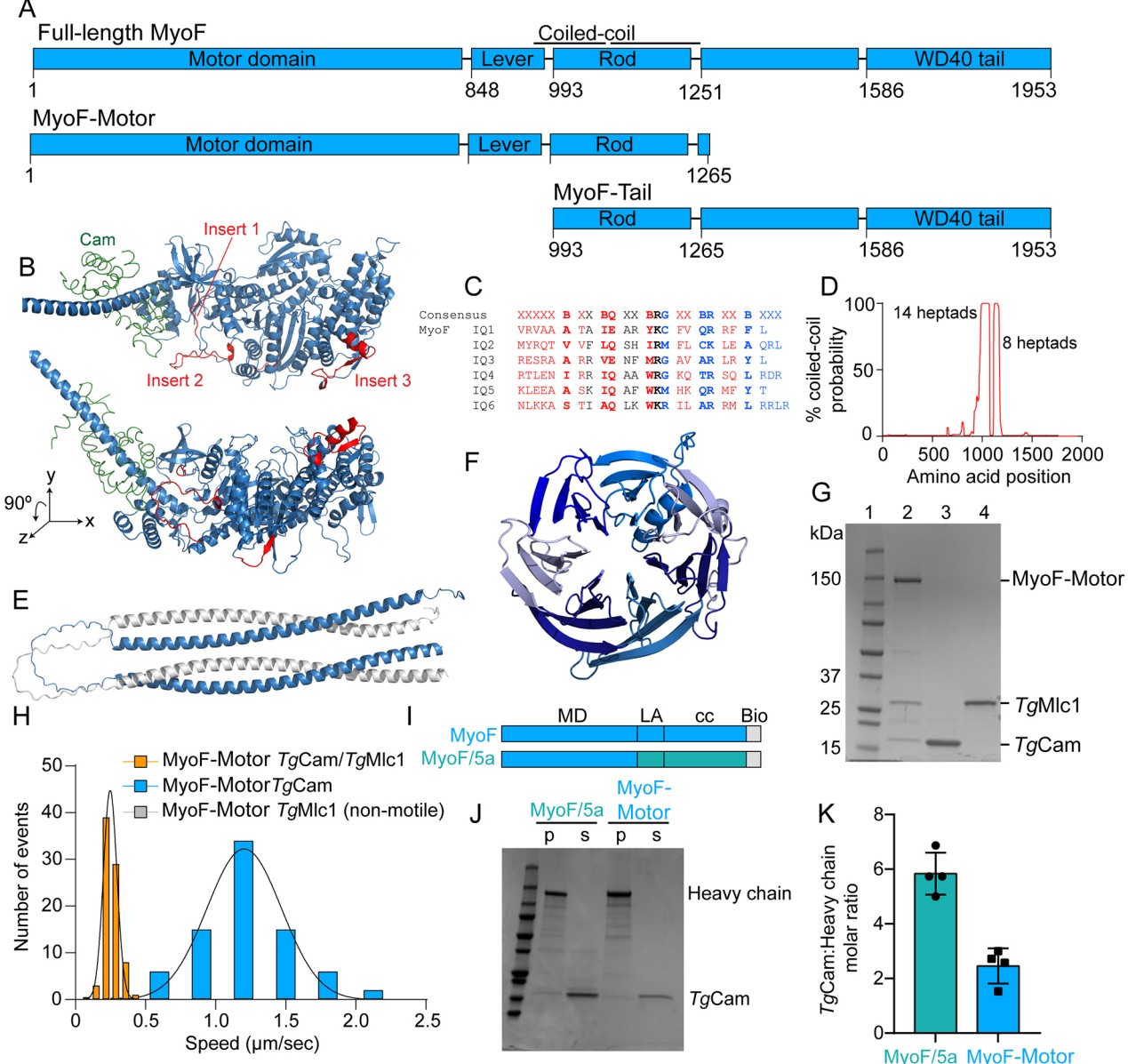

**Fig. 1. Analysis of MyoF domain organization and light chain binding.** (A) Domain organization of MyoF and constructs used in this study. Region predicted to form a coiled-coil is indicated. Amino acids 1175–1585 are sequence with unknown structure. (B) AlphaFold model of the MyoF motor domain (a.a. 1–847) and a portion of the lever arm (a.a. 848–885) (blue) showing three unique inserts (red). Calmodulin (green) (PDB 2DFS) is modeled onto the first IQ motif sequence. The structure presented beneath is rotated 90° clockwise about the *x*-axis. (C) Aligned sequences of the six predicted IQ motifs (consensus sequence IQxxxRGxxxR) in the MyoF lever arm (x, any residue; B, hydrophobic; red, C-lobe interacting; blue, N-lobe interacting). (D) Coiled-coil probability (Ncoils) of the MyoF sequence. Number of heptads are calculated from amino acid sequences with a *P*-value greater than 0.5 (50%) (I974–L1174). (E) Structure of the coiled-coil rod region of MyoF (E993–W1251) as predicted by AlphaFold. (F) Structure of the WD40 tail domain (a.a. M1572–A1941) as predicted by AlphaFold. Unstructured C-terminal amino acids G1942–V1953 are not shown. (G) Coomassie-stained SDS-PAGE gel of (lane 1) protein molecular mass marker; (lane 2) FLAG affinity purified MyoF-Motor from *Sf9* cells co-expressed with *Tg*Cam and *Tg*Mlc1; (lane 3) HIS purified *Tg*Cam and (lane 4) *Tg*Mlc1 expressed from *E. coli*. Also shown in Fig. S3A. Gel representative of four repeats. (H) Gliding filament *in vitro* motility speed distributions of MyoF-Motor bound to *Tg*Cam and *Tg*Mlc1 (orange, mean±s.d., 0.24±0.05 µm/s, *n*=80) and MyoF-Motor bound to only *Tg*Cam (blue, mean±s.d., 1.2±0.27 µm/s, *n*=78). (I) Domain organization of MyoF-Motor and MyoF/5a constructs showing MyoF sequences in blue and mammalian myosin 5a sequences in teal. (J) Coomassie-stained SDS-PAGE gel showing supernatant (s) and pellet (p) fractions of centrifuged heat-treated MyoF-Motor and MyoF/5a constructs. (K) Average calmodulin:heavy chain ratio for MyoF/5a (teal) and MyoF-Motor constructs (mean±s.d., *n*=3).

motif (Fig. 1B). The lever arm region of MyoF contains six putative light-chain binding IQ motifs (Fig. 1C). The dimerization domain of MyoF is predicted to begin within the last IQ motif and contains a total of 22 heptads with the potential to form an α-helical coiled-coil spanning amino acids (a.a.) I974 to L1174 (Fig. 1D). Although the ability of an α-helix to form a coiled-coil is dependent on the amino acid composition within the heptad repeat, 22 heptads should, in

principle, form a stable dimer, given that the first 20 heptads of vertebrate myosin 5a are sufficient for dimerization (Trybus, 2008). An AlphaFold model of the coiled-coil region (E993 to R1204) predicts an additional 77 amino acids of coiled-coil sequence, with a break in the middle of the rod region (Fig. 1E). This might prevent the rod from forming the extended conformation as is typical of some cargo transporting molecular motors, such as myosin 5a or

kinesin-1 (Vale, 2003; Hammer and Sellers, 2011). The coiled-coil region is followed by ∼400 amino acids of poorly predicted structure (see below), terminating with a seven-bladed WD40 β-propeller domain (Fig. 1F), the defining feature of this class of myosin motor. An alignment between *Tg*MyoF and MyoF motors from several related species indicate the domain organization is conserved; however, motor domain insert 1 is only conserved in the most closely related species including *Neospora* and *Eimeria*, whereas inserts 2 and 3 are present in all species but vary in both length and sequence. *Plasmodium* species contain an additional insert at the junction between the motor domain and lever arm that is not found in other species (Fig. S1).

Myosin light chains function to mechanically stabilize the lever arm, which is required for motility. Thus, to study the motile properties of recombinant MyoF *in vitro*, we first identified its native light chains by performing immunoprecipitation (IP) assays using GFP trap affinity resin and lysates from a parasite line where the endogenous protein was fused to a C-terminal EmeraldFP (EmFP) tag (Heaslip et al., 2016). A fraction of the eluted protein showed the presence of a ∼200 kDa band which likely corresponds to MyoF–EmFP, and bands of ∼60 and ∼15 kDa as shown by silver staining (Fig. S2A). MyoF-interacting proteins were subsequently identified using liquid chromatography-mass spectrometry (LC-MS) (Table S1). In two of the three IPs, two light chains were immunoprecipitated with MyoF: calmodulin (*Tg*Cam) (ToxoDB accession code TGME49_249240) and myosin light chain (*Tg*Mlc1) (ToxoDB accession code TGME49_257680) (Alvarez-Jarreta et al., 2024).

Having identified the native light chains, we next purified recombinant MyoF using the baculovirus/*Sf9* cell system. Because the tail domain of other classes of myosins interacts with the motor domain to form an auto-inhibited complex, we first purified a construct of MyoF lacking its WD40 tail domain (a.a. 1–1265), which was fused to a C-terminal biotin–FLAG tag (Fig. 1A, MyoF-Motor), with biotin and FLAG used for fluorescent labeling and affinity purification of the protein, respectively. The sequence of the biotin tag is derived from the biotin carboxyl carrier protein of acetyl-CoA carboxylase, which becomes biotinylated *in vivo* (Cronan, 1990; Li and Cronan, 1992). This construct was co-expressed in *Sf9* cells with untagged *Tg*Cam and *Tg*Mlc1 light chains in a bac-to-bac system, which ensures similar expression levels (Fig. S2B). All MyoF constructs in this study were also expressed bac-to-bac with a *T. gondii*-specific co-chaperone *Tg*UNC (ToxoDB accession code TGME49_249480) (Fig. S2B) (Bookwalter et al., 2014; Frenal et al., 2017). Although *Tg*UNC was not necessary for the expression of soluble MyoF, we found co-expression with the chaperone produced preparations with higher protein yields.

Using this expression strategy, we found that both *Tg*Cam and *Tg*Mlc1 co-purify with MyoF-Motor confirming their association with the motor (Fig. 1G, lane 2; Fig. S3A). To understand how *Tg*Cam and *Tg*Mlc1 differentially affect the activity of MyoF, we purified the motor co-expressed with *Tg*Cam alone (Fig. S3B) and with *Tg*Mlc1 alone (Fig. S3C). Using an *in vitro* motility assay, we found that complexes containing MyoF-Motor-*Tg*Cam-*Tg*Mlc1 moved filaments with an average speed of 0.24 µm/s (Fig. 1H; Movie 1), whereas the gliding motility speed for MyoF bound to only *Tg*Cam was enhanced fivefold (1.2 µm/s) (Fig. 1H; Movie 2). MyoF-Motor bound to only *Tg*Mlc1 was non-motile (Fig. 1H; Movie 3). These data suggest that *Tg*Cam is sufficient to fully occupy and mechanically stiffen the lever arm of MyoF to support motility, and that *Tg*Mlc1 might have a regulatory function. Sequence analysis shows that the lever arm region of MyoF has six putative light chain-binding sites (Fig. 1C). To determine whether all IQ motifs are capable of light chain binding, we counted the number of light chains using a calmodulin-binding assay described previously (Batters et al., 2012; Coluccio, 1994). MyoF-Motor bound to *Tg*Cam was thermally denatured, transferred to ice, and then centrifuged at high speed to separate soluble and insoluble fractions. Because the heavy chain precipitates and *Tg*Cam remains soluble, the concentration of the supernatant and pellet can be used to calculate the number of *Tg*Cam light chains bound per heavy chain. As a control, we purified a chimeric construct from *Sf9* cells containing the MyoF motor domain and mammalian myosin 5a lever arm and rod sequences (MyoF/5a), which is known to bind six calmodulin light chains (Fig. 1I; Fig. S3D). Using this strategy, we found that the chimeric MyoF/5a control binds approximately six calmodulins as expected (Fig. 1J,K, teal), whereas MyoF-Motor binds approximately three (Fig. 1J,K, blue) indicating that MyoF binds half as many light chains than myosin 5a and fewer than predicted from the amino acid sequence. When this experiment was performed with MyoF-Motor bound to *Tg*Mlc1 or *Tg*Cam and *Tg*Mlc1, *Tg*Mlc1 was predominately in the pellet precluding us from determining the number of *Tg*Mlc1 molecules that bind MyoF. However, when both *Tg*Mlc1 and *Tg*Cam are bound to MyoF, the amount of *Tg*Cam bound to MyoF decreases, suggesting that *Tg*Mlc1 displaces one or more *Tg*Cam molecules in order to bind MyoF (Fig. S3G).

To determine which IQ motifs are preferentially occupied by *Tg*Cam versus *Tg*Mlc1, we used AlphaFold to model the binding of each light chain to MyoF IQ motifs 1–3. The structural predictions indicated that both *Tg*Cam and *Tg*Mlc1 are capable of binding all three motifs (Fig. S4A). To assess potential binding preferences, we quantified light chain–IQ motif interactions by generating contact probability heatmaps and calculating interface scores based on minimum residue–residue distances (see Fig. S4 legend for details). These analyses revealed differential binding tendencies – *Tg*Cam was predicted to bind more strongly to IQ1 and IQ3, whereas *Tg*Mlc1 showed a higher predicted affinity for IQ2 (Fig. S4B).

Structurally, the reduced *Tg*Cam interaction with IQ2 appears to result from limited engagement of its N-terminal N-lobe, which adopts a more extended conformation in this complex (Fig. S4A). Conversely, *Tg*Mlc1 is predicted to interact with IQ2 with both lobes, leading to conformational differences that likely influence binding strength. Similar open or extended light chain conformations have been reported in other myosin–light chain complexes (Terrak et al., 2003).

Interface scores, calculated as the summed contact probabilities across the interaction surface further supported these predictions. Higher interface scores were observed for *Tg*Cam with IQ1 and IQ3, and for *Tg*Mlc1 with IQ2, consistent with the heatmap analysis (Fig. S4B). Taken together, these analyses suggest that *Tg*Mlc1 might preferentially occupy IQ2, potentially limiting *Tg*Cam binding at this site.

## A single MyoF dimer is non-processive

A hallmark of yeast and mammalian cargo transporting myosins is the ability to move processively on actin filaments as a single molecule (Hammer and Sellers, 2011). To determine whether a truncated dimer of MyoF is processive, we visualized the movement of MyoF-Motor on skeletal actin filaments bound to a quantum dot (Qdot). Streptavidin-coated Qdots were bound to MyoF-Motor through its C-terminal biotin tag at a ratio of 1 motor to 5 Qdots to ensure that the majority of Qdots are bound to a single motor. Under these mixing conditions, single molecules of MyoF-Motor bound to *Tg*Cam and *Tg*Mlc1 light chains did not move on actin (Movie 4).

Because light chain composition has a dramatic effect on the speed of MyoF-Motor in the *in vitro* gliding filament motility assay, we tested whether removing *Tg*Mlc1 had an effect on its processivity. We found that single molecules of MyoF-Motor bound to only *Tg*Cam also did not move on skeletal actin filaments (Movie 5). These results indicate that the single dimers of the truncated motor are unable to support motility and are therefore non-processive on skeletal actin.

We next tested whether small motor ensembles of MyoF-Motor bound to a single Qdot can move continuously on actin filaments. MyoF-Motor was mixed with Qdots at a 10:1 molar ratio to ensure saturation of streptavidin binding sites. Given its geometry and occupancy, this allows a single Qdot to bind 4-6 myosin motors (Hodges et al., 2009). We found that small teams of MyoF-Motor bound to *Tg*Cam and *Tg*Mlc1 were capable of supporting motility on skeletal actin filaments (Movie 6) with a characteristic run length (λ) of 0.65 μm (Fig. 2A, Table 1) and average speed of 0.51 μm/s (Fig. 2B, orange; Table 1). To test whether removal of *Tg*Mlc1 changes the properties of small ensembles of truncated MyoF, we imaged the movement of multiple MyoF-Motors on skeletal actin bound to only *Tg*Cam. We found little difference in the run length (0.72 μm, Table 1) of these ensembles compared to MyoF bound to both light chains (Movie 7; Fig. 2C). However, the average speed of movement was much faster (3.45 μm/s, Table 1), consistent with *in vitro* gliding filament motility results (Fig. 1H). Taken together, these data indicate that light chain composition affects the speed of MyoF but not the processivity or run length of the truncated MyoF-Motor construct.

### Teams of MyoF-Motor show optimal motility on actin-fascin bundles

Some classes of cargo transporting myosin motors have lever arms containing less than six IQ motifs and are optimized for movement on actin bundles rather than single actin filaments (Ropars et al., 2016; Nagy et al., 2008; Ricca and Rock, 2010). This property allows them to target bundled actin structures, such as filopodium and stereocilia, and become enriched at their tips (Vignjevic et al., 2006; Medalia et al., 2007; Berg and Cheney, 2002; Belyantseva et al., 2005; Delprat et al., 2005). To determine whether MyoF is targeted to actin bundles *in vivo*, we ectopically expressed the full-length motor in *Sf9* cells fused at its C-terminus with the mClover3 variant of GFP (MyoF–GFP) and imaged its localization using epifluorescence microscopy. We found that MyoF is strongly enriched to the ends of actin-enriched filopodium (Fig. S5).

Its ectopic localization and short (3IQ) lever arm length suggest that MyoF might move better on parallel actin bundles, which contain additional lateral binding sites that could enhance motility by ensuring at least one head remains bound to the actin. We compared the movement of teams of MyoF-Motor bound to a single Qdot on single actin filaments and actin-fascin, which crosslinks actin filaments into parallel bundles that are separated by 9 nm (Ishikawa et al., 2003). We found that teams of MyoF-Motor have a relatively low run frequency on single actin filaments (Fig. 3A, left, C). However, the frequency of movement and run length was dramatically enhanced on actin bundles (Fig. 3A, right, C,D, Table 1). The average speed of movement on bundles was reduced (Fig. 3E, Table 1), which might result from lateral movements and side stepping, a behavior seen with other molecular motors that are optimized to move on bundles (Sladewski et al., 2016). To test whether single molecules of MyoF-Motor move on bundles, we mixed clarified motors with Qdots at a 1:5 ratio to ensure only one motor is bound per Qdot and imaged its motility using total internal reflection fluorescence (TIRF) microscopy on actin-fascin bundles.

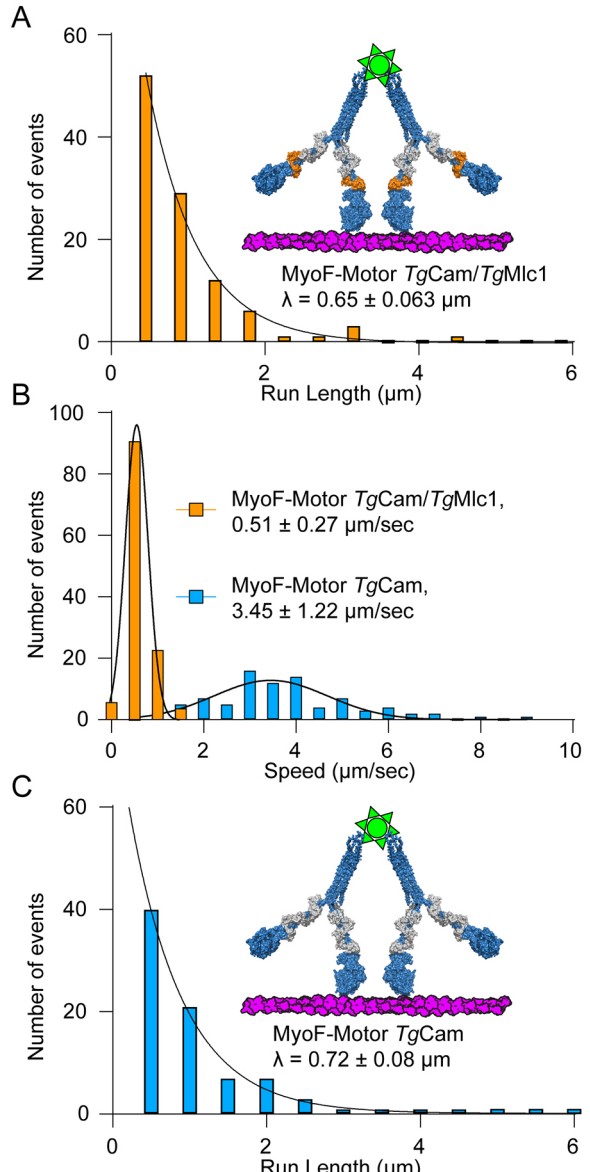

**Fig. 2. Effect of light chain composition on MyoF motility.** (A) Frequency distribution of run lengths for multiple motors of MyoF-Motor co-purified with *Tg*Cam and *Tg*Mlc1 bound to a Qdot on skeletal actin (*n*=124). Characteristic run length (λ) is shown ± standard error of the fitted parameter from nonlinear regression. (B) Histograms comparing multiple motor run speeds MyoF-Motor co-purified with *Tg*Cam and *Tg*Mlc1 (orange) (*n*=124) and MyoF-Motor co-purified with *Tg*Cam (blue) on phalloidin stabilized skeletal (Sk) actin (*n*=84). Histograms were fit to a single Gaussian distribution to determine the average speed as mean±s.d. for each construct. (C) Frequency distribution of run lengths for multiple motors of MyoF-Motor co-purified with *Tg*Cam on skeletal actin (*n*=124). Characteristic run length (λ) is shown ± standard error of the fitted parameter from nonlinear regression.

When we do this, no runs were observed (Movie 8) indicating that single molecules of MyoF-Motor are non-processive on actin bundles.

### The WD40 tail domain oligomerizes MyoF

We next used electron microscopy (EM) to determine the stoichiometry and organization of MyoF. Negatively stained EM images of the MyoF-Motor construct bound to *Tg*Cam were compared to the MyoF/5a chimera bound to *Mus musculus* calmodulin, which was mutated to prevent Ca$^{2+}$ binding

**Table 1. Summary of MyoF single molecule and multiple motor motility**

| Construct | No. of motors | Light chains | Actin | Velocity (µm/s, mean±s.d.) | Run length (λ) |
|---|---|---|---|---|---|
| **MyoF-Motor** | Single dimer | TgCam | Skeletal actin+Phalloidin | No movement | |
| | | TgMlc1 | | | |
| | | TgCam and TgMlc1 | | | |
| | Multiple dimers | TgCam | Skeletal actin+Phalloidin | 3.45±1.22 | 0.72±0.08 |
| | | TgMlc1 | | No movement | |
| | | TgCam and TgMlc1 | | 0.51±0.27 | 0.65±0.06 |
| | | TgCam and TgMlc1 | Skeletal actin+Phalloidin | 0.59±0.28 | 0.6±0.02 |
| | | TgCam and TgMlc1 | Skeletal actin+Phalloidin+Fascin | 0.28±0.17 | 1.8±0.24 |
| **FL-MyoF** | Single complexes | TgCam | Skeletal actin+Phalloidin | 0.9±0.88 | 1.1±0.85* |
| | | | Skeletal actin+Jas | 1.5±0.61 | 2.2±1.26* |
| | | | TgAct1+Jas | 1.5±0.53 | 1.14±0.19 |
| **MyoF/5a chimera** | Multiple dimers | CamΔall | Skeletal actin+Phalloidin | No movement | |
| | | | TgAct1+Jas | 2.6±1.14 | 0.45±0.05 |
| | Single dimer | | TgAct1+Jas | No movement | |

Run lengths are given as λ ± standard error of the fitted parameter from nonlinear regression expect for values highlighted by an asterisk, which are mean±s.e.m. See figure legends for detail of *n* values.

(CamΔall) (Krementsov et al., 2004), because the structure of the Myo5a backbone is well established (Trybus, 2008). MyoF-Motor shows 'V-shaped' particles containing pairs of 10 nm diameter heads consistent with a dimer (Fig. 4A, white arrowheads, blue). Compared to MyoF/5a, which has an extended lever arm (Fig. 4B, yellow arrowheads), the lever arm of MyoF was approximately half the length (Fig. 4A, yellow arrowheads; Fig. S6A), which is consistent with results showing that MyoF binds half as many calmodulin light

chains as a construct containing a myosin 5a lever arm (Fig. 1I–K). The C-terminal region of MyoF-Motor formed a globular structure (Fig. 4A, green arrowheads, red) rather than an extended coiled-coil rod like MyoF/5a (Fig. 4B, green arrowheads, red).

To investigate how the WD40 tail domain influences MyoF organization, we purified full-length MyoF containing a C-terminal biotin–FLAG tag co-expressed with TgCam (Fig. S2E) and visualized particles by negative-stain EM. The full-length protein

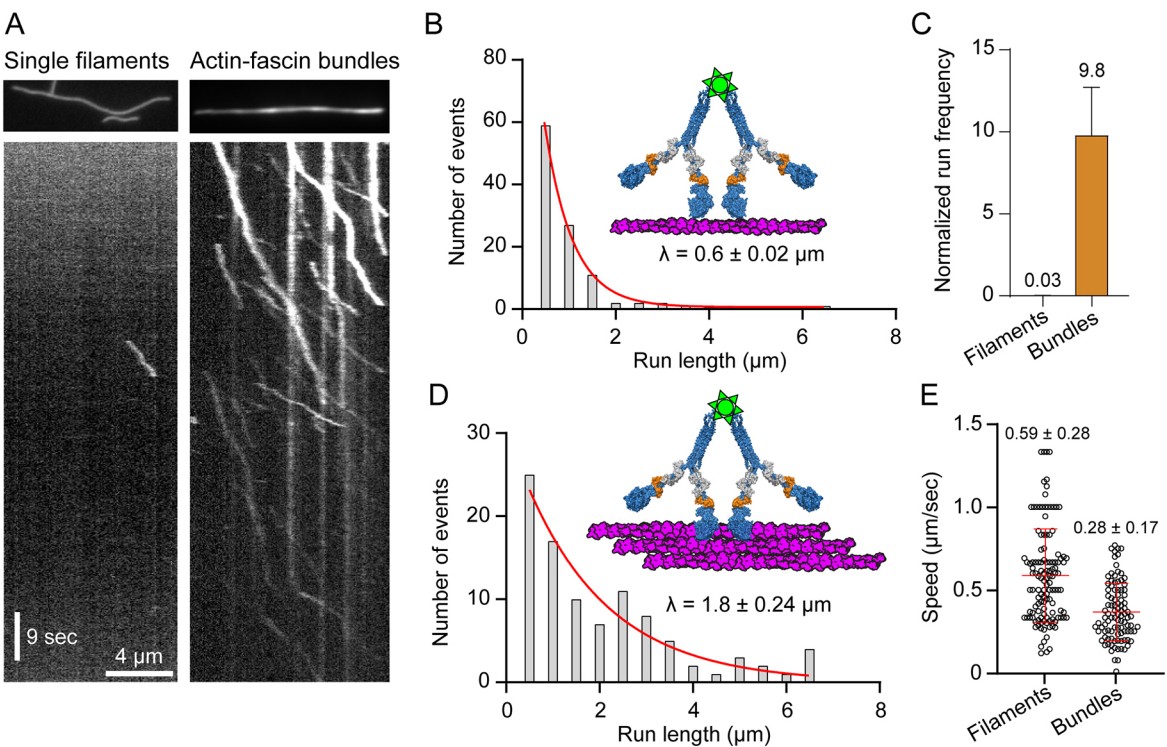

**Fig. 3. MyoF-Motor motility on single actin filaments and actin-fascin bundles.** (A) Kymograph (distance versus time) showing the movement (diagonal lines) of multiple motors of MyoF-Motor co-purified with TgCam and TgMlc1 on single skeletal actin filaments (left) versus actin-fascin bundles (right) at the same concentration of motor. (B) Frequency distribution of run lengths for multiple motors of MyoF-Motor on skeletal actin (*n*=124). Characteristic run length (λ) is shown ± standard error of the fitted parameter from nonlinear regression. (C) Comparison of the mean run frequency for multiple motors of MyoF-Motor on single skeletal actin filaments (mean±s.d., 0.03±0.01, *n*=22 filaments) versus MyoF-Motor on actin-fascin bundles (mean±s.d., 0.03±0.01, *n*=8 bundles). Event frequency was normalized per µM Qdot per µm actin track per s, as done previously (Sladewski et al., 2016). (D) Run length frequency distribution of multiple motors of MyoF-Motor on actin-fascin bundles (*n*=99). The characteristic run length (λ) on bundles is shown ± standard error of the fitted parameter from nonlinear regression and is significantly longer than on filaments (*P*<0.001, using the Kolmogorov–Smirnov Test). (E) Run speed distributions comparing multiple motors of MyoF-Motor on single actin filaments versus actin-fascin bundles. Mean±s.d. run speed is shown. Average run speeds are significantly different (*P*<0.0001, using a two-tailed unpaired *t*-test).

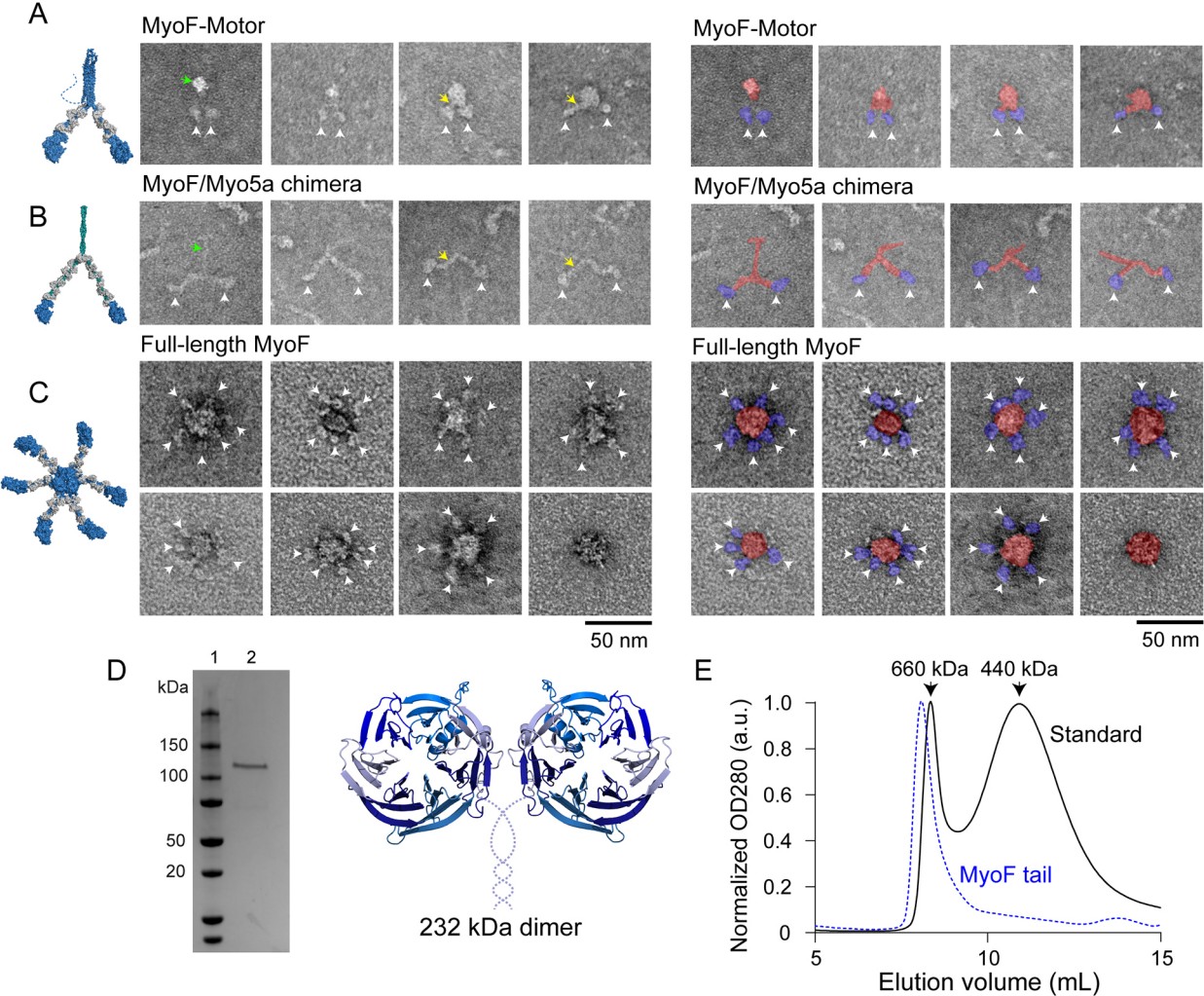

**Fig. 4. Electron micrographs and analytical SEC of MyoF constructs.** Montage of negatively stained images of (A) MyoF-Motor co-purified with *Tg*Cam showing 'v-shaped' particles typical of myosin dimers, (B) MyoF/5a chimeric construct co-purified with *Mus musculus* Cam which was mutated to prevent $Ca^{2+}$ binding (Cam∆all) (Krementsov et al., 2004), and (C) full-length MyoF co-purified with *Tg*Cam, which appears as a central density associated with up to 6 motor domains. White arrowheads indicate MyoF motor domains, which are paired in MyoF-Motor and MyoF/5a chimeric constructs. Yellow arrowheads indicate the lever arm, which is approximately half the length for MyoF compared to the lever arm of mammalian myosin 5a sequences. The green arrowhead points to the C-terminal dimerization domain which is compact for MyoF and an extended rod for MyoF/5a. Images are pseudo-colored (right) to indicate the motor domain (blue) and C-terminal regions (red). All images representative of two repeats. (D) Coomassie-stained SDS-PAGE gel (left) of the MyoF-Tail construct (a.a. 987–1953) purified from *Sf9* cells, fused to a C-terminal biotin–FLAG tag. Schematic depiction (right) of the MyoF-Tail construct represented as a 232 kDa dimer. Dotted lined indicates sequence of unknown structure. Gel representative of three repeats. (E) Analytical SEC showing the elution of 440 kDa (ferritin, 10.98 ml peak) and 660 kDa (thyroglobulin, 8.37 ml peak) standards (black) compared to the elution of MyoF-Tail construct (blue, 8.18 ml peak) indicating that the WD40 tail domain oligomerizes into a complex that is at least hexameric. Results representative of two repeats.

exhibited a central circular density, with multiple peripheral densities consistent with myosin motor domains. Quantification of visible heads revealed that most particles contained three to six heads (Fig. 4C; Fig. S6B). Because negative-stain EM often underestimates the number of observable heads due to particle orientation effects, these results might be consistent with a hexameric organization of the full-length MyoF complex.

We next tested whether the WD40 tail domain of MyoF is capable of forming a higher order oligomer. To do this, we purified a MyoF tail construct containing the coiled-coil region, linker and WD40 domain (a.a. 987–1953) (MyoF-Tail) (Figs 1A and 4D). When applied to a calibrated size exclusion chromatography (SEC) column, the protein eluted before the 660 kDa standard (Fig. 4E). Because the molecular mass of a MyoF-Tail dimer is 232 kDa, this indicates that the oligomerization state is at least hexameric (696 kDa). Because a protein of this molecular mass would elute

in the void volume, it is not possible to distinguish a hexamer from a higher order oligomer. Taken together, these data indicate that the WD40 tail changes the stoichiometry from a two-headed dimer into a larger complex containing at least six heads.

To gain insight into how the WD40 tail domain might facilitate oligomerization, we used AlphaFold to determine whether the MyoF W40 domain could form a hexameric arrangement. The model predicts that pairs of WD40 domains form dimers stabilized by extensive hydrogen bonding networks (Fig. S7A–D, orange, E), positioning the N-terminal regions of each tail adjacent to one another. These dimers can further organize into a trimer of dimers through additional inter-dimer hydrogen bonding, consistent with a hexameric assembly (Fig. S7A–D, green, F; Movie 9). Detailed analysis of the interfaces reveals that the same sets of amino acid residues mediate contacts across all three dimer interfaces, with an additional hydrogen bonding network stabilizing the trimeric assembly.

Given this predicted hexameric arrangement of the WD40 domains, we next investigated the positioning of the region between the coiled-coil and WD40 domains. AlphaFold predictions of the hexameric coiled-coil and mid region preceding the WD40 domain show that the coiled-coil forms an α-helical bundle (Fig. S8, gray), and residues 1252–1464 of the mid region form a series of short α-helices that associate with this bundle (Fig. S8, blue). However, the C-terminal portion of the mid region (residues P1465–A1571) is unstructured and predicted with low confidence, as indicated by pLDDT scores and low overall pTM and ipTM scores (Fig. S8, yellow). Together, these predictions support a model in which the coiled-coil region and the mid region positions the WD40 domains to form the circular density observed in negative-stain EM of the full-length MyoF, although the exact organization of the preceding mid region remains poorly defined.

### Full-length MyoF is processive on *Tg*Act1 filaments and skeletal actin bundles

Because full-length MyoF oligomerizes into a multi-headed complex, we first asked whether a single hexameric complex could move processively on actin filaments. To test this, we imaged individual full-length MyoF complexes co-purified with *Tg*Cam moving on rabbit skeletal actin filaments stabilized with Alexa Fluor 488–phalloidin. Motors were labeled with Alexa Fluor 647–streptavidin at a 1:5 molar ratio to ensure the fluorescent dye bound to a single complex. Under these conditions, full-length MyoF did not move processively on skeletal actin filaments (Movie 10).

We next examined MyoF motility on *Tg*Act1 filaments. Alexa Fluor 647–streptavidin-labeled full-length MyoF was introduced into flow chambers containing jasplakinolide-stabilized *Tg*Act1 filaments. In contrast to skeletal actin, single MyoF complexes moved robustly and processively on *Tg*Act1 (Fig. 5A; Movie 11), with a characteristic run length (λ) of 1.14 μm and an average velocity of 1.5 μm/s (Fig. 5B,C, Table 1).

Structural studies have shown that actin-stabilizing agents trap actin in distinct conformational states (Pospich et al., 2020). Our recent work has demonstrated that, unlike skeletal actin, unstabilized *Tg*Act1 filaments adopt an open D-loop conformation, and that jasplakinolide acts to further stabilize this open conformation (Hvorecny et al., 2024) which might promote MyoF binding and processivity. Thus, to determine whether jasplakinolide itself was required for processivity, we imaged single MyoF complexes on unstabilized *Tg*Act1 filaments labeled with the actin chromobody. MyoF remained processive under these conditions (Movie 12), indicating that jasplakinolide does not artificially promote processivity and that the intrinsic structural features of *Tg*Act1 support MyoF motility.

Interestingly, we frequently observed MyoF pausing at filament ends prior to dissociation (Fig. 5D,E). This behavior is atypical for cargo-transporting myosins and might provide insight into its cellular function. Analysis of motile events show that when full-length MyoF reached the end of filaments the mean±s.e.m. pause time was 1.5±0.26 s (Fig. S9, blue).

To determine how *Tg*Mlc1 influences MyoF behavior, we purified full-length MyoF containing a C-terminal biotin tag co-expressed with both *Tg*Cam and *Tg*Mlc1 (Fig. S2F). In the presence of both light chains, MyoF frequently accumulated at filament ends but did not exhibit processive motility on *Tg*Act1 (Movie 13), suggesting that *Tg*Mlc1 modulates motor activity by reducing processivity.

Finally, because the MyoF-Motor construct exhibited enhanced movement on actin–fascin bundles, we tested whether full-length MyoF complexes could move on bundled actin. This experiment was performed with skeletal actin because fascin could not bind

*Tg*Act1 and no *Tg*Act1 bundling factors have been identified to date. On phalloidin-stabilized bundles, motors primarily exhibited static associations, with only occasional movement characterized by frequent pausing, slow velocities, and short run lengths (Fig. 5F–H, left, Table 1). In contrast, on jasplakinolide-stabilized bundles, full-length MyoF moved efficiently with minimal pausing, higher velocities, and longer run lengths compared to single *Tg*Act1 filaments (Fig. 5F–H, right; Table 1; Movie 14).

To gain deeper mechanistic insight into why full-length MyoF is processive on *Tg*Act1 filaments and not skeletal actin filaments, we compared the movement of a MyoF/5a chimeric construct (which contains the MyoF motor domain and myosin 5a backbone) (Fig. 1I) on skeletal and *Tg*Act1 actin isoforms. The use of chimeric myosin constructs was previously used by Krementsova et al. to define the structural and functional features that contribute to processive movement of class 5 myosins (Krementsova et al., 2006). We found that multiple MyoF/5a motors bound to a Qdot did not move on skeletal actin (Movie 15) but runs were observed on jasplakinolide-stabilized *Tg*Act1 actin filaments (Movie 16). The characteristic run length (λ=0.45 μm) of MyoF/5a (Fig. S10A; Table 1) was much shorter than that of full-length MyoF (Fig. 5B, Table 1), and the average speed of movement was similar to MyoF-Motor bound to *Tg*Cam (2.6 μm/s) (Fig. S10B; Table 1). We also found that unlike the full-length motor, the MyoF/5a dimer was non-processive as a single motor on *Tg*Act1 filaments (Movie 17). This result indicates that the kinetic and thermodynamic properties of the MyoF motor domain likely requires oligomerization into a hexameric organization to support processivity.

### MyoF binds microtubules via its tail domain and localizes to intraconoid microtubules *in vivo*

We previously have shown that MyoF localizes to both the cytosol and the parasite periphery, with the cytosolic pool controlling the dynamics of cytoplasmic actin and the peripheral pool driving a peripheral actin flow (Periz et al., 2019; Kellermeier and Heaslip, 2024). First, we sought to determine how MyoF controls cytoplasmic actin organization. One hypothesis, motivated by the structural similarity between the MyoF WD40 domain and other actin-binding proteins (coronin and Aip1), is that the MyoF tail might bind actin filaments and control actin organization by crosslinking actin filaments via its WD40 and motor domains. To investigate this, we tested whether the MyoF tail binds actin filaments. Recombinantly purified MyoF tail was labeled with Alexa Fluor 488–streptavidin and was added to a flow cell bound with Rhodamine–phalloidin stabilized skeletal actin filaments. Epifluorescence microscopy showed no association of the tail with single actin filaments, even under low salt conditions (Fig. S11). Thus, in light of our evidence showing that the MyoF dimer further oligomerizes through the WD40 tail domain, this result supports an alternative model where MyoF drives cytosolic actin dynamics by interacting with multiple filaments simultaneously through its N-terminal head domains rather than the tail to translocate filaments relative to one another.

Next, we sought to understand how MyoF is anchored to the parasite periphery. We find that tubulin consistently copurifies with full-length MyoF from both *Sf9* and *T. gondii* lysates (Fig. S2E,F; Table S1). *T. gondii* has several tubulin-based structures including 22 subpellicular microtubules that run approximately two-thirds the length of the parasite, the conoid made up of 14 tubulin fibers, and two intraconoid microtubules (Fig. 6A) (Morrissette and Sibley, 2002; Liu et al., 2016). To test whether MyoF associates with any tubulin-based structures, we used ultrastructure expansion microscopy (U-ExM) to determine the precise peripheral

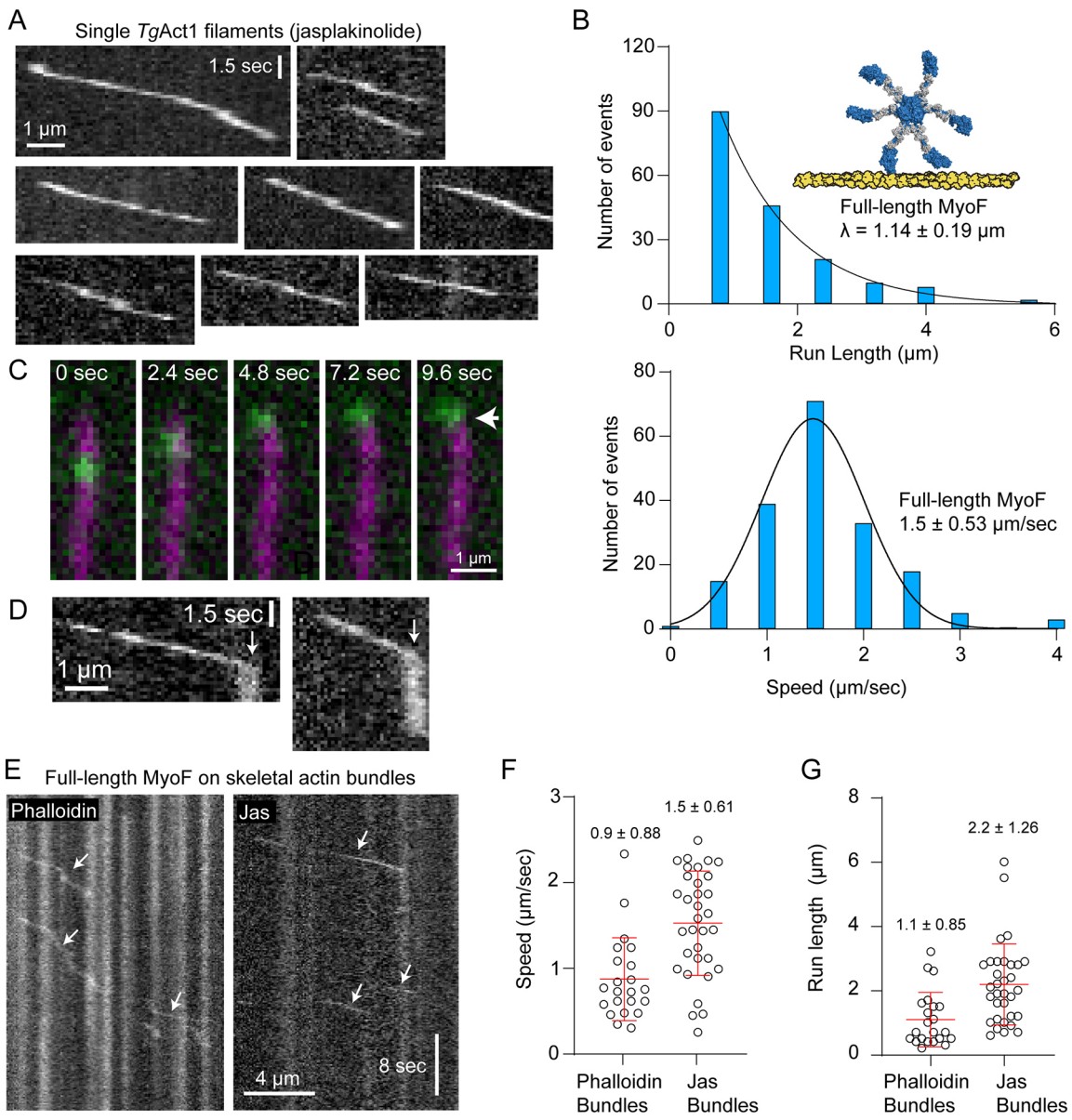

**Fig. 5. Motile properties of full-length MyoF.** (A) Kymographs (distance versus time) of single molecules of full-length MyoF labeled with Alexa Fluor 647–streptavidin moving on jasplakinolide-stabilized *Tg*Act1 filaments. (B) Run length distribution (top) [characteristic run length distribution (λ) shown ± standard error of the fitted parameter from nonlinear regression] and speed distribution (bottom) (mean±s.d. shown) of full-length MyoF bound to *Tg*Cam moving on *Tg*Act1 filaments (*n*=187). (C) Montage showing full-length MyoF bound to Alexa Fluor 647–streptavidin (green) moving to the end of a *Tg*Act1 filament (magenta, white arrow). The *Tg*Act1 filament was imaged using the chromobody fused to EmeraldFP. (D) Kymographs of MyoF showing movement on a *Tg*Act1 filament (diagonal line), followed by retention at the end of filaments (vertical line, white arrow). (E) Kymographs of single molecules of full-length MyoF bound to *Tg*Cam labeled with Alexa Fluor 647–streptavidin moving on actin-fascin bundles made from skeletal actin stabilized with (left) phalloidin or (right) jasplakinolide. Arrows highlight directed runs. (F) Speed (mean±s.d.) and (G) run length (mean±s.d.) of actin-fascin bundles from skeletal actin stabilized with either phalloidin or jasplakinolide. Jasplakinolide bundles, *n*=33; phalloidin bundles, *n*=22. Average run speeds are significantly different (*P* value<0.0001, using a two-tailed unpaired *t*-test). Average run lengths are significantly different (*P*=0.0008).

localization of MyoF–GFP. Visualization of MyoF–GFP and acetylated tubulin, a marker for subpellicular microtubules and the conoid show that MyoF localized to discrete puncta along the parasite periphery that did not co-localize with subpellicular microtubules (Fig. 6B, magnification b), suggesting association with another pellicular component, potentially within the inner membrane complex (IMC) (Back et al., 2024). Notably, MyoF also appeared as a filamentous structure within the center of the conoid (Fig. 6B, magnification a), consistent with localization along the intraconoid microtubules.

**MyoF translocates actin along microtubules**

To determine whether MyoF binds microtubules directly, we labeled the full-length motor with Alexa Fluor 647–streptavidin and added it to flow chambers containing fluorescently labeled Taxol-stabilized microtubules. We found that full-length MyoF bound to microtubules at nanomolar concentrations (Fig. 6C). We also found that the MyoF-Tail construct labeled with Alexa Fluor 647–streptavidin also binds microtubules, indicating that the C-terminus of MyoF is sufficient for microtubule association (Fig. 6D). Timelapse movies indicate that the binding for both constructs is static, suggesting that the binding is

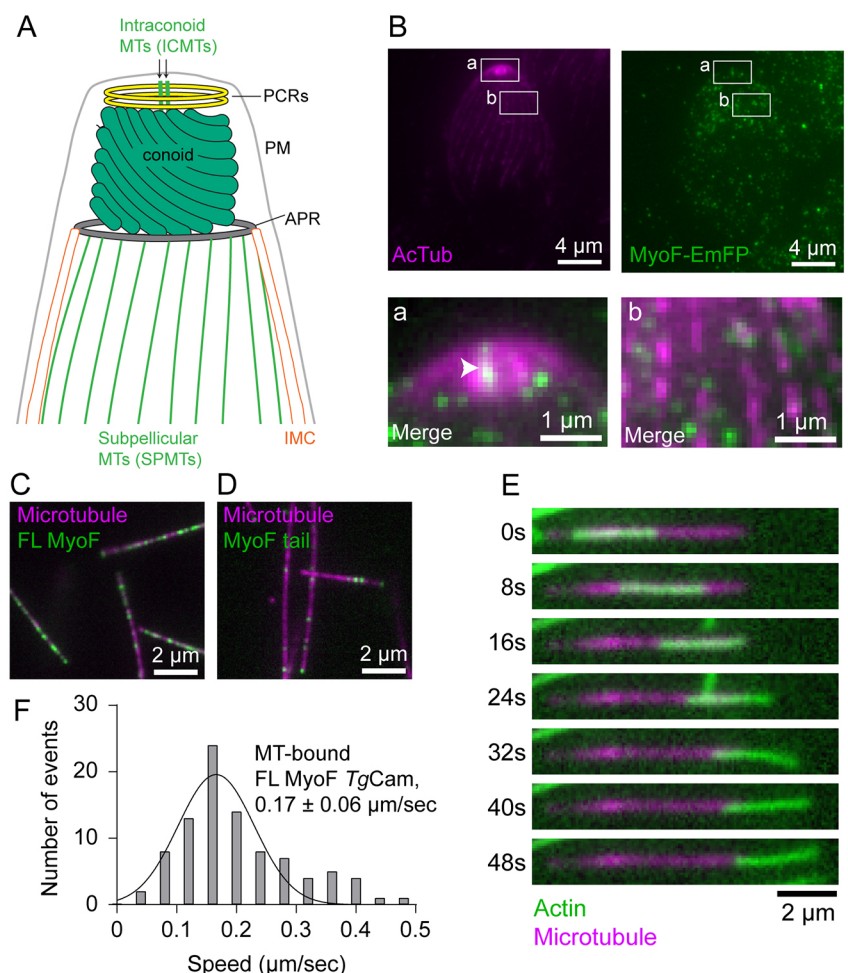

**Fig. 6. MyoF localizes to the IMC and ICMTs and can translocate actin along microtubules.** (A) Schematic showing the cytoskeletal organization of the conoid, subpellicular microtubules, inner membrane complex (IMC), apical polar ring (APR), plasma membrane (PM) preconoidal rings (PCRs, yellow) and intraconoid microtubules (ICMTs, green) in *T. gondii*. (B) U-ExM showing the localization of acetylated tubulin (magenta) and MyoF–GFP (green). Magnifications of the indicated areas are shown below highlighting the filamentous localization of MyoF in the center of the conoid (a, arrowhead) and to the IMC (b). Images are single slices of deconvolved images. Brightness and contrast of insets was adjusted to show SPMTs. (C,D) Epifluorescence microscopy image showing that full-length MyoF–Alexa647 (C, green) and MyoF-Tail-Alexa647 (D, green) bind statically to microtubules (magenta). (E) Montage showing the movement of a skeletal actin filament (green) along a MyoF-bound microtubule (magenta). Images in B and E are representative of two repeats; images in C and D are representative of three repeats. (F) Speed distribution of actin motility on MyoF bound microtubules (mean±s.d., 0.17±0.06. n=94).

likely not a consequence of an electrostatic charge–charge interaction between the motor and microtubule E-hooks (Movies 18, 19). Given that the MyoF WD40 tail domain interacts directly with microtubules, we considered a model in which MyoF anchors to microtubules while its N-terminal motor domains engage actin filaments. To test this, we assembled microtubules decorated with MyoF and introduced fluorescent actin filaments. Strikingly, actin filaments underwent directed translocation along MyoF-bound microtubules (Movie 20) with an average speed of 0.17 μm/s (Fig. 6E,F). Similar to the end retention on actin observed in single-molecule assays (Fig. 5G,H), filaments were often found to be retained at their plus (barbed)-end before dissociating (Fig. 6E). To determine whether the interaction between MyoF and microtubules persists at physiological ionic strength, we repeated the assay in the presence of 150 mM KCl. MyoF-mediated translocation was maintained under these conditions, indicating that MyoF is capable of associating with microtubules at physiological ionic strength (Movie 21).

Together, these findings support a model in which MyoF associates with microtubule-based structures in the parasite and may coordinate interactions between the actin and microtubule cytoskeletons.

## DISCUSSION

Myosin F is a class 27 myosin motor that is unique to apicomplexan parasites and protozoa in the alveolate group of eukaryotes (Mueller et al., 2017; Kellermeier and Heaslip, 2024; Odronitz and Kollmar, 2007). Loss of MyoF in *T. gondii* results in defects in the organization and dynamics of the actin cytoskeleton, which in turn

disrupts vesicle transport, organelle inheritance during cell division and organization of the endomembrane pathway (Heaslip et al., 2016; Venugopal et al., 2020; Carmeille et al., 2021). To understand how MyoF functions in this process, we studied the properties of this motor which are completely undefined.

## MyoF forms an oligomer using its WD40 tail domain

The defining feature of MyoF is the presence of a WD40 tail domain (Mueller et al., 2017). Approximately 1% of proteins in the human genome contain WD40 domains, and they are thus found in proteins with diverse cellular functions and often act as scaffolds for the formation of large protein complexes (Stirnimann et al., 2010; Xu and Min, 2011). The WD40 domains in the actin-binding proteins coronin and Aip1 bind to actin directly, so we hypothesized that MyoF might regulate actin organization by crosslinking actin filaments via its motor domain and the WD40 tail domain as several other molecular motors regulate the organization of their respective cytoskeletal tracks in this manner (Tang et al., 2016; Laplante et al., 2015). However, MyoF does not regulate actin organization in this way because the WD40 domain does not bind to single actin filaments. Rather, our negatively stained images show that full-length MyoF forms an oligomer consisting of a central hub formed by its WD40 domains. Emanating from this, we visualized up to six densities that were the size of myosin motor domain. Given that the coiled-coil domain of MyoF can induce dimerization of the MyoF-Motor construct, this suggests that full-length MyoF forms a hexamer, or trimer of dimers. Although speculative, particles with

fewer than six visible heads could reflect autoinhibitory head–tail interactions that promote a compact conformation, thereby obscuring one or more motor domains in negative-stain EM. Our AlphaFold prediction of the WD40 domain supports a hexameric assembly organized as a trimer of dimers, stabilized by extensive and symmetric hydrogen-bonding networks across conserved interfaces (Fig. S7). In this model, the N-termini of the WD40 domains are arranged as three adjacent pairs, positioning the upstream coiled-coil and motor domains in close proximity. Such an architecture would be well suited to facilitate intramolecular interactions between heads and the tail domain or neighboring subunits.

Consistent with this idea, the predicted structure of the coiled-coil and mid-region preceding the WD40 domain suggests that these regions associate with the central helical bundle but exhibit low confidence in the C-terminal portion (Fig. S8), potentially reflecting conformational flexibility. This flexibility might permit dynamic rearrangements between extended and compact states. A compact, autoinhibited conformation, analogous to those described for other myosins, could reduce the spatial separation of motor domains, leading to partial masking or steric occlusion in projection images. Thus, the observation of particles with three to five visible heads might not necessarily indicate sub-stoichiometric assembly but instead reflect conformational heterogeneity within a hexameric complex.

Oligomerization of WD40 domains has been reported previously in several systems. Multiple WD40-containing proteins have been shown to form dimers (Stirnimann et al., 2010; Asano et al., 2001), and the WD40 domains of DDB1 assembles into a trimeric architecture (Li et al., 2006). In these examples, the WD40 domain primarily contributes to scaffold formation within larger multiprotein complexes rather than directly driving higher-order self-assembly. To our knowledge, our data represents the first example in which a WD40 domain appears to mediate stable homo-oligomerization of its parent protein, forming a defined hexameric assembly.

Based on these data, we propose a model in which MyoF assembles into a multi-motor domain complex capable of engaging adjacent actin filaments and sliding them relative to one another, thereby regulating actin organization and dynamics within the cell (Fig. 7, expansion 1). In this model, oligomerization enables coordinated motor activity across multiple filaments, promoting filament rearrangement rather than simple cargo transport. This mechanism is conceptually analogous to the tetrameric kinesin Eg5 (also known as KIF11), which crosslinks and slides antiparallel microtubules within the mitotic spindle using motor domains positioned at opposing ends of the molecule (Kapoor et al., 2000). Consistent with this model, loss of MyoF results in altered actin organization and dynamics in vivo (Kellermeier and Heaslip, 2024), supporting a role for MyoF as an active regulator of actin architecture.

## MyoF WD40 domain binds microtubules

In cells, MyoF localizes to the parasite periphery, where we hypothesize it drives peripheral actin flow and promotes movement of dense granules adjacent to the pellicle (Heaslip et al., 2016; Periz et al., 2019). No colocalization between MyoF and the subpellicular microtubules was observed, suggesting that peripheral anchoring is instead mediated by an as-yet-unidentified IMC-associated component (Fig. 7, expansion 2). In contrast, MyoF exhibits a distinct apical localization consistent with the intraconoid microtubules (ICMTs) (Hu et al., 2002) (Fig. 6B). These ~350 nm microtubules traverse the center of the hollow conoid and are implicated in positioning the rhoptries within the conoid (Dolinsky et al., 2004). The ICMTs are also associated with several microtubule-associated vesicles (MVs) of unknown function. Given the established role of MyoF in transporting multiple vesicle populations, including dense granules and Rab6- and ROP1-positive vesicles, it will be important to determine whether MyoF similarly contributes to positioning or transport of rhoptries or MVs within the conoid (Fig. 7, expansion 3). The association between MyoF and ICMTs is likely direct, as both full-length and tail constructs bind robustly to taxol-stabilized brain microtubules at nanomolar concentrations.

Two regions of the MyoF tail may contribute to microtubule binding. First, the disordered region predicted by AlphaFold (Fig. S8; residues P1465–A1571), which exhibits low pLDDT confidence, contains multiple R/K pairs and is strongly basic (predicted pI=11.38), consistent with electrostatic microtubule-binding mechanisms described for other microtubule-associated proteins (Ludzia et al., 2021; El Mammeri et al., 2022). Alternatively, electrostatic surface analysis of the hexameric WD40 domain reveals a continuous electropositive surface spanning multiple β-propeller domains on two faces of the assembly (Fig. S12). This extensive basic surface is a candidate for the microtubule-binding interface.

Through direct interaction with microtubules, full-length MyoF could potentially crosslink and coordinate actin filaments relative to

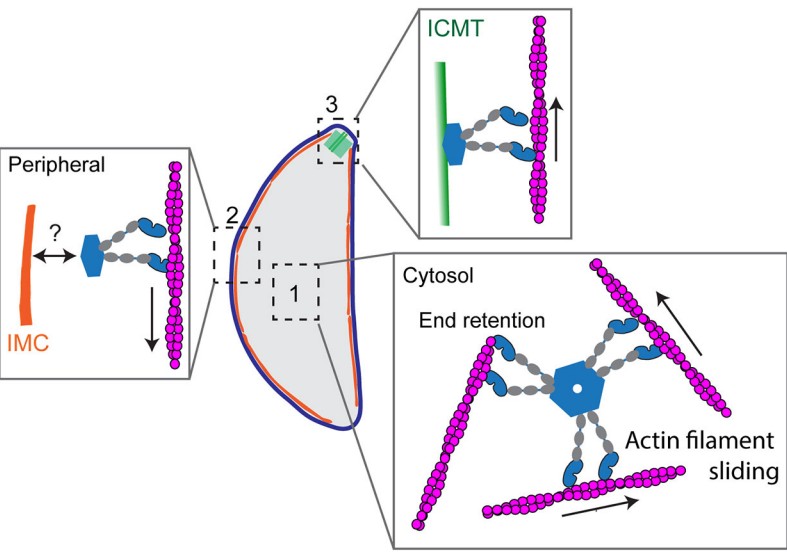

**Fig. 7. Proposed model of MyoF function.** MyoF has three subcellular localizations. Expansion 1, hexameric MyoF organizes cytoplasmic actin filaments. Multiple pairs of motor domains interact with and translocate actin filaments thereby driving their dynamics in the cell. Expansion 2, MyoF associates with the IMC through an unknown binding adapter. Expansion 3, MyoF localizes to the intraconoid microtubules. The function of MyoF at the conoid requires further investigation. The oligomeric state of MyoF at the IMC and ICMTs unknown (in expansions 2 and 3) and is drawn as a dimer for simplicity.

the microtubule cytoskeleton. Thus, IMC- and ICMT-associated MyoF might regulate actin filament organization and dynamics within distinct cellular compartments.

### MyoF motility is influenced by the composition of the actin track

Single molecules of MyoF-Motor (dimers) and full-length MyoF (oligomers) are non-processive on phalloidin-stabilized single filaments skeletal actin. Although small ensembles of MyoF-Motors move inefficiently (i.e. short run lengths and a low run frequency). In contrast, full-length MyoF moved processively on jasplakinolide-stabilized single *Toxoplasma* filaments and exhibited longer run-lengths and faster speeds on jasplakinolide-stabilized skeletal bundles compared with phalloidin stabilized bundles (Fig. 5A–C) indicating that motor activity is influenced by the structural states of F-actin. While Toxoplasma and skeletal actin have similar overall helical parameters, we previously found an altered D-loop confirmation in *Toxoplasma* actin compared to skeletal actin (Hvorecny et al., 2024). Jasplakinolide- and phalloidin-stabilized skeletal actin exhibit an 'open' D-loop confirmation (Pospich et al., 2020). As the motile properties of other molecular motors are influenced by changes at the D-loop (Kubota et al., 2009), it is tempting to speculate that it is this confirmation that confers track specificity of MyoF, however future structural studies will be required to define the basis of the motors altered activity. Taken together, these data indicate that the enhanced motility on *Tg*Act1, which shares only 83% identity to skeletal actin filaments, results from a preference of the MyoF motor domain with its native track. Because a two-headed MyoF construct is non-processive on *Tg*Act1 filaments, the duty cycle of MyoF is insufficient to support motility of a two-headed MyoF construct on *Tg*Act1 filaments. Thus, oligomerization by the MyoF-Tail is necessary to form an approximately six-headed complex that supports single-molecule motility of the full-length motor.

MyoF motility was enhanced on actin bundles compared with single filaments, with an ∼300-fold increase in run frequency (Fig. 3C). In addition, full-length accumulates at the tips of filopodia in *Sf9* cells (Fig. S3). Several classes of myosin motors are optimized for movement on bundles including Myo5c (Sladewski et al., 2016) and Myo10 (Ropars et al., 2016; Nagy et al., 2008; Caporizzo et al., 2018), although the actin bundle preference of these motors is conferred via different mechanisms. Myo5c forms a parallel dimer through its coiled-coil domain and additional flexibility in the lever arms accommodates bundle motility (Sladewski et al., 2016), whereas Myo10 forms an anti-parallel coiled-coil so the orientation of the heads is optimized for movement on adjacent filaments on actin bundles in filopodia (Ricca and Rock, 2010). In both these cases, motors encounter actin filament bundles in cells (Geron et al., 2013; Jacobs et al., 2009). The physiological implications of enhanced motility of MyoF on bundles requires further investigation because it is not known whether *Toxoplasma* actin forms bundled networks in cells. The high turnover rate of *T. gondii* actin (Hvorecny et al., 2024) precludes imaging in fixed cells (Kellermeier and Heaslip, 2024) and makes ultrastructural imaging a challenge. In addition, *T. gondii* does not contain a fascin homolog. In fact, no actin-bundling proteins have been identified in *T. gondii* to date, so whether MyoF encounters parallel actin filament bundles in cells remains to be determined.

### Myosin light chain composition regulates MyoF velocity

MyoF contains six identifiable IQ motifs based on primary amino acid sequence. However, negative stain EM and a calmodulin-binding assay revealed that MyoF has a short lever arm that binds only three light chains. This is reminiscent of other myosins where there is a mismatch between the number of IQ motifs and number of associated light chains. *Leishmania* myosin-21 has 16 predicted light chain binding IQ motifs but only binds a single calmodulin (Batters et al., 2012), and the third IQ sequence in myosin 15 does not bind a light chain, collectively indicating that presence of an identifiable IQ motif is not a reliable predictor of light chain binding (Bird et al., 2014).

We used immunoprecipitation to identify *Tg*Cam and *Tg*Mlc1 as the light chains associated with MyoF. *Tg*Cam binding is consistent with a previous report that identified MyoF as a close interacting partner of *Tg*Cam (Song et al., 2022). *Tg*Mlc1 is a known binding partner of another parasite specific myosin (MyoA) involved in parasite motility and invasion (Meissner et al., 2002). Data from the gliding filament assays (Fig. 1) and myosin motility assays (Fig. 2) showed that MyoF velocities is dependent on light chain composition. Although we cannot exclude the possibility that other myosin light chains bind to MyoF, our data indicates that the lever arm is fully occupied in the presence of *Tg*Cam because MyoF-Motor bound to *Tg*Cam supports *in vitro* gliding motility. When in the presence of *Tg*Mlc1 alone MyoF is non-motile indicating incomplete occupancy. When binding both *Tg*Cam and *Tg*Mlc1, MyoF velocity is ∼6-fold slower than when the motor is bound to *Tg*Cam alone indicating that *Tg*Mlc1 serves a regulatory function. Given that MyoF binds three *Tg*Cam molecules instead of six, as predicted by the number of identifiable IQ motifs, future structural studies are needed to resolve which IQ motifs are occupied by light chains and to determine the structure of the remaining unoccupied IQ motifs. As shown for Myo10, which contains three IQ motifs following by a single α-helix positioned in a manner to favor movement on bundles (Ropars et al., 2016), structural determination of this region of MyoF will shed light on the orientation of the lever arm in relation to the motor domain would provide insight into MyoF's preference for bundled actin.

Despite the importance of the actin cytoskeleton to *T. gondii* growth and pathogenicity, the proteins and mechanisms that regulate actin organization are poorly understood. Characterizing the unique domain organization of MyoF and biophysical properties revealed a new myosin-based mechanism of actin organization and dynamics. Given the absence of canonical proteins that control the formation of bundled and branched actin networks in *T. gondii* and related organisms, motor-driven actin organization might represent an evolutionary divergent mechanism of actin regulation.

## MATERIALS AND METHODS
### Cell and *T. gondii* culture

*T. gondii* tachyzoites derived from RH strain were used in this paper. Construction of the MyoF–EmFP-expressing parasites was published previously (Heaslip et al., 2016). Parasites were maintained by continuous passage in human foreskin fibroblasts purchased from ATCC (HFFs) in Dulbecco's modified Eagle's medium (DMEM; Thermo Fisher Scientific) containing 1% (v/v) heat inactivated fetal bovine serum (FBS; VWR, Radnor PA, USA) and 1× antibiotic/antimycotic (Thermo Fisher Scientific).

### Identification of MyoF-associated light chains

To identify MyoF-associated light chains, ∼$10^9$ extracellular MyoF–EmFP parasites (Heaslip et al., 2016) were lysed in 10 ml of GFP-trap buffer (25 mM imidazole pH 7.4, 300 mM KCl, 1 mM EGTA, 0.5% Triton X-100, 2 mM DTT and 1:100 protease inhibitor cocktail (Millipore Sigma; cat. #P8340) on ice for 10 min. Lysates were precleared at 16,000 *g* for 20 min at 4°C. 25 µl of GFP-Trap (ProteinTech, cat. #GTMA) were resuspended and then washed twice in GFP-trap buffer. The cleared lysates were added to the magnetic beads and incubated with gentle shaking for 1 h

at 4°C. Using a magnetic rack, beads were washed ten times in GFP-trap buffer. After the final wash beads were wash twice in 1× PBS and resuspended in a final volume of 100 µl. 10 µl of beads were removed and added to an equal volume of 2× SDS–PAGE loading buffer (0.12 M Tris-HCl pH 6.8, 0.14 M SDS, 5% glycerol, 0.15 mM bromophenol blue, 0.6 M β-mercaptoethanol and 5 mM DTT) boiled for 5 min at 95°C. The remaining beads were stored in 1× PBS. LC-MS/MS was performed at the University of Connecticut Proteomics and Metabolic Facility.

## Constructs

The *Tg*MyoF (ToxoDB: TgME49_278870) coding sequence was cloned via PCR using RH cDNA. MyoF was then subcloned into a pFastBac expression vector so that the protein would be expressed in frame with a C-terminal FLAG tag used for affinity purification followed by a biotin tag for conjugation to fluorescent streptavidin reporters (Fig. S2B). The 88-amino-acid biotin tag is derived from the *Escherichia coli* biotin carboxyl carrier protein which becomes biotinylated at a single lysine residue (Cronan, 1990; Li and Cronan, 1992). For purification from bacteria, *Tg*Mlc1 containing an N-terminal 6×HIS tag was cloned into the pET3a bacterial expression vector (Novagen) and *Tg*Cam containing a C-terminal 6×HIS tag was cloned into the pET22b bacterial expression vector (Novagen). For co-expression in *Sf9* cells, untagged *Tg*Mlc1 and *Tg*Cam were cloned into pFastBac (Thermo Fisher Scientific) for baculovirus expression. The sequence of *Homo sapiens* FSCN1 (UniProt accession Q16658) was codon optimized for *E. coli* and fused at its N-terminus with a HAT-mNeonGreen sequence for HIS affinity purification and visualization using fluorescence microscopy. A TEV cleavage site was inserted between mNeon and FSC1 to generate untagged fascin. This sequence was inserted into pET16b (Novagen) for expression in BL21 (DE3) *E. coli*. Protein structural models were computed using AlphaFold3 (Abramson et al., 2024) using the MyoF protein sequence from ToxoDB.org, accession number TgME49_278870 as input (Alvarez-Jarreta et al., 2024). Coiled coil prediction was performed using Ncoils (https://bio.tools/ncoils)

## Protein expression and purification

*T. gondii* light chains *Tg*Cam and *Tg*Mlc1 were fused to a 6×HIS tag for nickel affinity purification from BL21 (DE3) bacterial cells. Volumes of 500 ml Luria Broth (LB) were inoculated with a 25 ml starter culture of BL21 (DE3) transformed with *Tg*Cam or *Tg*Mlc1 light chain expression constructs and grown to an optical density of ~0.8–1.2. Cells were then chilled, induced for protein expression with the addition of 0.4 mM isopropyl β-D-1-thiogalactopyranoside (IPTG), and incubated in a 16°C shaking incubator overnight. Cells were then harvested by centrifugation (700 *g* for 10 min) and frozen at −20°C. Cell pellets were thawed and resuspended in 50 ml ice-cold lysis buffer [10 mM NaPO$_4$, 300 mM NaCl, 0.5% glycerol, 7% sucrose, 0.5 mM PMSF, 0.5 mM DTT, 0.5% NP-40, protease inhibitors (Pierce cat. #A32953), pH 7.4] and lysed by incubating with 3 mg/ml lysozyme (Sigma, cat. #L6876) on a 4°C rocking incubator for 30 min. Lysed cells were then sonicated and centrifuged for 35 min at 250,000 *g*. Supernatant was added to 1 ml prepared nickel select affinity resin (Sigma, cat. #P6611) equilibrated with 0.5 M imidazole followed by wash buffer to remove the imidazole and batch incubated at 4°C on a rocking platform for 45 min. The lysate was then applied to a chromatography column and washed with 60 column volumes of HIS wash buffer (10 mM NaPO$_4$, 300 mM NaCl, 0.5% glycerol and 10 mM imidazole, pH 7.4), followed by 20 column volumes HIS wash buffer containing 10 mM imidazole. Bound protein was then eluted with elution buffer (10 mM NaPO$_4$, 300 mM NaCl, 0.5% glycerol, 200 mM imidazole, pH 7.4, with 0.5 mM DTT) in 1 ml fractions. Fractions that were positive for protein by Bradford were pooled, concentrated to ~2 ml using Amicon centrifugal filters and dialyzed against storage buffer (25 mM imidazole, pH 7.4, 300 mM NaCl, 50% glycerol, 1 mM DTT, 1 µg/ml leupeptin) overnight at 4°C.

Human fascin was purified similarly give light chains except, following elution from nickel select affinity resin, ~10 mg of protein was dialyzed into TEV cleavage buffer (10 mM phosphate, pH 7.4, 50 mM NaCl, 1 mM DTT), clarified at 400,000 *g* for 15 min at 4°C and cleaved TEV protease (NEB P8112S) was used for 1 h at 27°C. NaCl was then added to the mixture at a final concentration of 250 mM and 0.25 ml of nickel select affinity resin

equilibrated in 0.5 M imidazole, followed by TEV binding buffer was added and incubated on a nutating rocker at 4°C for 1 h. The mixture was then centrifuged (14,000 *g* for 5 min) and the supernatant containing untagged fascin was dialyzed in storage buffer (25 mM imidazole, pH 7.4, 300 mM NaCl, 50% glycerol, 1 mM DTT, 1 µg/ml leupeptin) overnight at 4°C.

Myosin constructs were co-expressed with the indicated light chain(s) in *Sf9* cells using the baculovirus system. Infected *Sf9* cells were incubated for 72 h at 27°C, harvested by centrifugation (700 *g* for 10 min) and resuspended in 60 ml FLAG lysis buffer {10 mM imidazole, pH 7.4, 0.3 M NaCl, 5 mM MgCl$_2$, 7% sucrose, 1 mM EGTA, 1 mM DTT, 2 mM MgATP, 25 µg/ml indicated light chain and protease inhibitors ([1× Sigma (cat. #P8340), 0.5 mM PMFS, 5 µg/ml leupeptin (Sigma cat. #L2884, and 1 mg/ml benzamidine]}. Cells were lysed by sonication and then clarified by centrifugation for 35 min at 250,000 *g*. The supernatant was added to 2 ml FLAG affinity resin (Thermo Fisher Scientific, cat. #PIA36803) equilibrated in FLAG wash buffer (10 mM imidazole, pH 7.4, 0.3 M NaCl, and 1 mM EGTA) and incubated in batch for 45 min on a rocking shaker at 4°C. The FLAG resin was then collected by gentle centrifugation (5 min at 800 rpm) and applied to a column for washing with ~60 ml FLAG wash buffer at 4°C. Bound protein was then eluted with 0.1 mg/ml FLAG peptide in wash buffer. Fractions positive for protein by Bradford were pooled and concentrated to 0.5 ml using a 10 kDa MWCO Amicon Ultra Centrifugal Filter (Millipore, cat. #2022-08-01) and applied to a Superdex 200 Increase 10/300 GL (GE Healthcare, cat. #28990944) gel filtration column equilibrated in FLAG wash buffer. Fractions corresponding to full-length MyoF were pooled, concentrated to ~1.5 ml, and dialyzed against storage buffer (25 mM imidazole, pH 7.4, 300 mM NaCl, 50% glycerol, 1 mM DTT, 1 µg/ml leupeptin) overnight at 4°C.

Skeletal actin, *T. gondii* actin, actin chromobody (actin CB-EmFP) and NEM myosin were prepared as described previously (Hvorecny et al., 2024). Kinesin406 G235A ('rigor' kinesin) was prepared as described previously (Sladewski et al., 2023).

## Thermal denaturation assay

*Sf9*-purified myosin (20 µg) co-purified with the indicated light chain were mixed with 50 µl of B50 buffer (50 mM KCl, 25 mM imidazole, pH 7.4, 1 mM EGTA, 4 mM MgCl$_2$, and 10 mM DTT). Samples were boiled for 2 min and immediately transferred to ice. An aliquot (10 µl) was reserved for protein concentration determination by Bradford assay and for SDS-PAGE analysis. The remaining 40 µl was centrifuged at 84,000 *g* for 20 min at 4°C using a TLA100 rotor. Following centrifugation, the supernatant was collected and its concentration determined. The supernatant was then mixed with 2× SDS-PAGE loading buffer and boiled. The pellet was resuspended directly in 50 µl of hot 2× SDS-PAGE loading buffer. Both the supernatant and pellet fractions (30 µl of each) were resolved by SDS-PAGE to confirm that only free *Tg*Cam was present in the supernatant. The molar ratio of light chain to heavy chain was calculated by subtracting the molar amount of *Tg*Cam or calmodulin detected in the supernatant fraction from the total molar amount of *Tg*Cam or calmodulin present in the boiled reserved 'total' sample.

## *In vitro* gliding filament assay

*In vitro* motility flow chambers were coated with a nitrocellulose film and adsorbed with biotinylated bovine serum albumin (Bio-BSA; Thermo Fisher Scientific) by incubating with 0.5 mg/ml Bio-BSA for 1 minute in buffer B150 (150 mM KCl, 25 mM imidazole, pH 7.4, 1 mM EGTA, 4 mM MgCl$_2$ and 10 mM DTT). Flow cells were then washed and blocked with 0.5 mg/ml BSA in buffer B150 for 1 min, washed and functionalized with neutravidin by adding 50 µg/ml neutravidin (Thermo Fisher Scientific) in buffer B150 for 1 min followed by three washes with buffer B150 to remove unbound neutravidin. MyoF constructs were prepared by performing an initial actin spindown to remove motors that are unable to dissociate from actin in the presence of MgATP. To do this, MyoF constructs were diluted to 0.7 µM in buffer B150. To this 1.4 µM skeletal F-actin and 10 mM MgATP was added, mixed and centrifuged for 20 min at 350,000 *g*. The soluble fraction was then applied to the neutravidin functionalized flow chamber for 2 min and washed with buffer B150 to remove unbound motor and MgATP. The flow chamber was then infused with 50 nM rhodamine-phalloidin-labeled actin

for 1.5 min and washed with buffer B (50 mM KCl, 25 mM imidazole, pH 7.4, 1 mM EGTA, 4 mM MgCl$_2$, and 10 mM DTT). Gliding motility was initiated with the addition of *in vitro* motility Go buffer [50 mM KCl, 25 mM imidazole, pH 7.4, 1 mM EGTA, 4 mM MgCl$_2$, 0.25% methylcellulose, 10 mM DTT, 0.1 mg/ml of the indicated light chain, 2 mM MgATP, an ATP regeneration system (1 mM phosphocreatine, 0.1 mg/ml creatine phosphokinase) and oxygen scavenging system (50 µg/ml catalase, 3 mg/ml glucose, and 125 µg/ml glucose oxidase)]. Gliding filaments were imaged at 37°C in epifluorescence using a DeltaVision Elite microscope (Cytiva) built on an Olympus base with a 100×1.39 NA objective and definite focus system. Images were acquired every 1 s for 2 m with a scientific CMOS camera and DV Insight solid-state illumination module. Filament motility was analyzed using the FIJI plug-in MTrackJ plugin (Schindelin et al., 2012).

### Single and multiple motor motility assays

For imaging the motility of single molecules and small ensembles of MyoF-Motor constructs, the motor was diluted in buffer M (10 mM imidazole pH 7.4, 4 mM MgCl$_2$, 1 mM EGTA, 300 mM KCl) and clarified 400,000 *g* for 15 min at 4°C, and its concentration was determined by a Bradford assay. For single-molecule experiments, 80 nM MyoF-Motor was mixed with 400 nM Qdot 655 streptavidin conjugate (1:5 ratio) in buffer M containing 4 mg/ml BSA and 0.5% Pluronic F-127. At this ratio, the majority of Qdots are bound to a single motor. For experiments visualizing the motility of small ensembles of MyoF-Motor constructs, 1 µM MyoF-Motor was mixed with 0.1 µM Qdot 655 streptavidin conjugate (10:1 ratio) in the same buffer, which promotes the binding of multiple motors per Qdot. For the attachment of stabilized F-actin filaments, NEM-treated muscle myosin was absorbed to the surface of a flow chamber, washed with buffer B (50 mM KCl, 25 mM imidazole, pH 7.4, 1 mM EGTA, 4 mM MgCl$_2$ and 10 mM DTT) and blocked with blocking buffer (50 mM KCl, 25 mM imidazole, pH 7.4, 1 mM EGTA, 4 mM MgCl$_2$, 4 mg/ml BSA, 0.5% Pluronic F-127, and 10 mM DTT). The flow chamber was then rinsed with buffer B and infused with 50 nM rhodamine phalloidin or Alexa Fluor 488–phalloidin-stabilized skeletal F-actin filaments. After 2 min, the flow chamber was rinsed with buffer B to remove unbound filaments, and myosin mixtures were diluted 1:50 to 1:200 into Go buffer [50 mM KCl, 25 mM imidazole, pH 7.4, 1 mM EGTA, 4 mM MgCl$_2$, 10 mM DTT, 4 mg/ml BSA, 0.5% Pluronic F-127, 2 mg/ml kappa casein, 0.1 mg/ml of the indicated light chain, 2 mM MgATP, and oxygen scavenging system (50 µg/ml catalase, 3 mg/ml glucose and 125 µg/ml glucose oxidase)], added to the flow chamber and imaged at 37°C with an imaging rate of 5 to 20 frames per seconds where indicated. Images were acquired using epifluorescence microscopy on a DeltaVision Elite microscope or using TIRF microscopy on a Nikon AXR built around the inverted Ti2E microscope and captured on Photometrics Prime 95B sCMOS camera where indicated.

Single-molecule experiments with full-length MyoF were done similarly except MyoF was prepared by diluting into B500 (500 mM KCl, 25 mM imidazole, pH 7.4, 1 mM EGTA, 4 mM MgCl$_2$, and 10 mM DTT) and centrifuged at 400,000 *g* for 15 min at 4°C. The motor was then diluted to 300 nM and added to Alexa Fluor 647–streptavidin conjugate that was diluted into B500 and clarified 400,000 *g* for 15 min at 4°C at a molar ratio of 1:5. This mixing ratio ensures that the majority of streptavidin fluorophores are bound to a single motor. The motility of single full-length motors was visualized on Alexa Fluor 488–phalloidin-stabilized skeletal F-actin bound to partially PEGylated flow cells (Sladewski et al., 2018). For single and ensemble experiments using *T. gondii* actin filaments, *T. gondii* actin filaments were stabilized with jasplakinolide at a molar ratio of 1:1.2 and attached to the surfaces of flow cells similarly to stabilized skeletal actin and imaged in Go buffer supplemented with 1 µM jasplakinolide and 50 nM actin chromobody-EmeraldFP for visualization. For experiments where full-length MyoF motility was observed on unstabilized *Tg*Act1 filaments, 110 µM *Tg*Act1 filaments were polymerized for 30 min with the addition of 1× polymerization buffer (25 mM imidazole, pH 7.4, 4 mM MgCl$_2$ and 0.1 mM MgATP) and incubated for 30 min at 37°C. The polymerized actin was then diluted 100-fold and added to NEM-myosin-coated flow chambers for 2 min and unbound actin was washed out. Single molecules of MyoF labeled with Alexa Fluor 647–streptavidin conjugate were prepared as described above for imaging single molecule motility. Under conditions where *Tg*Act1 filaments

are unstabilized, imaging was performed within 20 minutes after actin was diluted.

### Dynamic actin–microtubule crosslinking assay

Labeled microtubules were prepared by polymerizing a mixture of cycled bovine tubulin and Cy5 tubulin (PurSolutions) followed by stabilization with paclitaxel (Cytoskeleton, Denver, USA) as described previously (Sladewski et al., 2023). To attach microtubules, kinesin406 G235A ('rigor' kinesin) was diluted to 0.1 mg/ml in buffer B and added to a partially PEGylated flow chamber for 2 min. The flow chamber was then washed with buffer B, blocked with blocking buffer for 2 min, and rinsed with buffer B. The flow chamber was then infused with 0.9 µM taxol stabilized Cy5-labeled microtubules for 2 min and then rinsed with buffer B containing 10 µM paclitaxel to remove unbound microtubules. Unlabeled clarified full-length MyoF was diluted to 10 nM in actin/MT Go buffer [80 mM KCl, 25 mM imidazole, pH 7.4, 1 mM EGTA, 4 mM MgCl$_2$, 10 mM DTT, 4 mg/ml BSA, 0.5% Pluronic F-127, 2 mg/ml kappa casein, 0.1 mg/ml *Tg*Cam, 0.25% methylcellulose, 10 µM paclitaxel, 2 mM MgATP, and oxygen scavenging system (50 µg/ml catalase, 3 mg/ml glucose, and 125 µg/ml glucose oxidase)] and added to the flow chamber for 5 min to allow the MyoF-Tail to bind microtubules. The flow cell was then rinsed with buffer B containing 10 µM paclitaxel to remove unbound MyoF and then actin/MT Go buffer containing 0.1 µM Alexa Fluor 488–phalloidin-stabilized skeletal F-actin was added to the flow chamber which was subsequently sealed using nail polish. Movement of actin along microtubules was observed at 37°C using epifluorescence microscopy as described above and images were captured every 10 s for 20 min.

### MyoF-Tail–microtubule binding assay

The MyoF-Tail (a.a. 987–1953) containing a C-terminal biotin–FLAG tag was diluted in buffer M, clarified at 400,000 *g* for 15 min at 4°C, and its concentration was determined by a Bradford assay. The protein was diluted to 0.1 µM and mixed with 0.5 µM pre-clarified Streptavidin–Alexa Fluor 488 conjugate (Thermo Fisher Scientific) to ensure binding of a single molecule per fluorophore. Partially PEGylated flow chambers were functionalized, passivated, and bound to fluorescently labeled microtubules as described above. The labeled MyoF-Tail was diluted 1:25 in modified Go buffer [50 mM KCl, 25 mM imidazole, pH 7.4, 1 mM EGTA, 4 mM MgCl$_2$, 10 mM DTT, 4 mg/ml BSA, 0.5% Pluronic F-127, 2 mg/ml kappa casein, 10 µM paclitaxel, 1 mM MgATP, and oxygen scavenging system (50 µg/ml catalase, 3 mg/ml glucose, and 125 µg/ml glucose oxidase)], added to the flow chamber for 6 min, washed with modified Go buffer, and the microtubules and labeled MyoF were imaged using epifluorescence microscopy on a DeltaVision Elite microscope.

### MyoF-Tail and full-length MyoF-actin binding assay

Rhodamine–phalloidin-stabilized skeletal actin adhered flow chambers were prepared as described above. MyoF-Tail in buffer M was and clarified 400,000 *g* for 15 min at 4°C, and its concentration was determined by a Bradford assay. The protein was then diluted to 1 µM and mixed with 0.1 Qdot 655 streptavidin conjugate (5:1 ratio) to promote the binding of multiple molecules per Qdot. The mixture was then diluted 1:50 in a modified low-salt imaging buffer [10 mM KOAc, 25 mM imidazole, pH 7.4, 1 mM EGTA, 4 mM MgCl$_2$, 10 mM DTT, 0.5% Pluronic F-127, 2 mg/ml kappa casein, 2 mM MgATP, and oxygen scavenging system (50 µg/ml catalase, 3 mg/ml glucose, and 125 µg/ml glucose oxidase)], added to the flow chamber, and imaged using epifluorescence microscopy on a DeltaVision Elite microscope.

### Ectopic expression in *Sf9* cells

$20×10^6$ *Sf9* cells grown to mid-log phase were infected with 1 ml of a baculovirus encoding full-length MyoF fused at the C-terminus the mClover3 variant of GFP and incubated 2.5 h on a rocking platform at room temperature. Cells–virus mixture was then added to 5 ml of *Sf9* medium, and 2 ml was added to a MatTek dish and allowed to infect for 2 days at 25°C. Medium was aspirated and 2 ml of cold 4% paraformaldehyde in 1× PBS was added to the well and fixed for 20 min. The well was then washed three times with 1× PBS, permeabilized with 0.25% Triton X-100, washed three times with 1× PBS,

and 1× PBS containing 20 µM rhodamine–phalloidin (Thermo Fisher Scientific) was added and incubated on a nutator for 1 h. The well was washed three times with 1× PBS and imaged in TRITC/FITC channels using epifluorescence microscopy on a DeltaVision Elite microscope.

## Expansion microscopy

Human foreskin fibroblasts (HFFs) were grown to confluency on 12 mm round coverslips in a 12-well plate and infected overnight with *T. gondii* parasites expressing endogenous levels of MyoF fused at the C-terminus to emeraldFP (MyoF–EmFP) (Heaslip et al., 2016). The following day, the cells were fixed in 4% paraformaldehyde at room temperature for 20 min, washed three times in 1× PBS, transferred to a 6-well plate containing FA/AA solution (1x PBS, 1.2% paraformaldehyde and 2% acrylamide), sealed with parafilm, and incubated at 37°C overnight. The next day, the FA/AA solution was removed, and cells were washed once with PBS. To induce polymerization, 90 µl monomer/gel solution (19% sodium acrylate, 10% acrylamide, 0.1% BIS and 1.1× PBS) was mixed with 5 µl of 10% TEMED and 5 µl of 10% ammonium persulfate (APS). 35 µl of this solution was added to parafilm in a 6 well plate stored on ice. The coverslip was then immediately inverted onto the gel solution and allowed to polymerize for 5 min and then transferred to a 37°C incubator for 1 h. Afterwards, 2 ml of denaturation buffer (50 mM Tris-HCl pH 9, 200 mM SDS and 200 mM NaCl) were added to the wells and placed on a rocker for 15 min at room temperature. Using forceps and a razer, the gel was detached from the coverslip and transferred to a 1.5 ml Eppendorf tube containing 1.5 ml of denaturation buffer and incubated for 90 min at 95°C. Gels were then transferred to a Petri dish containing 25 ml of deionized water and incubated for 30 min at room temperature. The expended gels were then washed twice with water, quartered with a razor and washed again with 1× PBS. One segment of the gel was transferred to a 12-well dish containing blocking buffer (2% BSA, 1× PBS and 0.5% Tween-20) and rocked on a shaker at room temperature for 30 min. The gel slice was treated with primary antibody [1:500 mouse anti-acetylated tubulin (Sigma T6793) and 1:500 rabbit anti-GFP-Alexa Fluor 488 (Invitrogen A21311)] diluted in blocking buffer and rocked overnight at room temperature. The next day, the gel was washed three times in 2 ml wash buffer (1× PBS and 0.5% Tween-20) and bound to secondary antibody [1:500 goat anti-mouse-IgG Alexa Fluor 647 (Thermo Fisher Scientific, A-21235) or 1:500 goat anti-rabbit-IgG Alexa Fluor 488 (Thermo Fisher A-11008)] for 2.5 h. The gel slice was then washed three times with 2 ml wash buffer followed by three times with distilled water incubating 30 min between each wash. The expanded gel was then transferred to a poly-D-lysine-coated MatTek dish and imaged using epifluorescence microscopy on a DeltaVision Elite microscope. The expansion factor was determined by measuring the diameter of the gel before and after expansion.

## Analytical size exclusion chromatography

MyoF-Tail and protein standards were dialyzed into gel filtration buffer [10 mM imidazole, pH 7.4, 300 mM NaCl, 1 mM EGTA, 1 µg/ml leupeptin and 1 mM DTT]. Samples (400 µl) containing either 0.25 mg/ml MyoF-Tail or 0.5 mg/ml protein standards (see Fig. 4E legend) were clarified by centrifugation at 30,000 *g* for 20 min at 4°C and loaded separately onto a Superdex 10/300 GL size-exclusion column equilibrated in the same buffer.

## Data availability, resources and AI usage statement

Plasmids and cell lines are freely available upon request. Python scripts for generation of TgMyoF structural models were written with the help of ChatGPT and are available in Zenodo (doi:10.5281/zenodo.19116060). After using these services, the authors reviewed and edited the content as needed and take full responsibility for the content of the publication.

## Acknowledgements

We thank Drs Jeremy Balsbaugh and Jennifer Liddle at UConn Proteomics and Metabolomics for their assistance with the mass spectrometry analysis, Dr Chris O'Connell at the UConn Advanced Microscopy Facility for assistance with TIRF microscopy, Dr Michael Griffith (University of Connecticut), Dr Sabina Absalon (Indiana University), Dr Christopher de Graffenried (Brown University), and Dr Paul Campbell for assistance with the expansion microscopy experiments.

## Competing interests

The authors declare no competing or financial interests.

## Author contributions

Conceptualization: T.E.S., A.T.H.; Data curation: T.E.S.; Formal analysis: T.E.S.; Funding acquisition: A.T.H.; Investigation: T.E.S., A.T.H.; Validation: T.E.S.; Writing – original draft: T.E.S.; Writing – review & editing: T.E.S., A.T.H.

## Funding

This work was supported by the National Institute of General Medical Science R35GM138316 awarded to A.T.H., R35GM138316-02S1 awarded to A.T.H. and T.E.S., and National Institute of Allergy and Infectious Disease grant R21AI185834 awarded to T.E.S. Open Access funding provided by National Institute of General Medical Sciences. Deposited in PMC for immediate release.

## First person

This article has an associated First Person interview with the first author of the paper.

## Data and resource availability

Plasmids and cell lines are available upon request. Python scripts for generation of TgMyoF structural models are available in Zenodo (doi:10.5281/zenodo.19116060). All other relevant data and details of resources can be found within the article and its supplementary information.

## Peer review history

The peer review history is available online at https://journals.biologists.com/jcs/lookup/doi/10.1242/jcs.264520.reviewer-comments.pdf

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
