## [Peer Review File · Journal of Cell Science]

Molecular features of myosin F adapted for driving actin flows in *Toxoplasma gondii*

Thomas E. Sladewski and Aoife T. Heaslip

DOI: 10.1242/jcs.264520

Editor: Michael Way

Review timeline

Original submission:	20 October 2025
Editorial decision:	10 December 2025
First revision received:	1 March 2026
Accepted:	2 March 2026

Original submission

First decision letter

MS ID#: jcs.264520

MS TITLE: Molecular features of Myosin F adapted for driving actin flows in *Toxoplasma gondii*

AUTHORS: Thomas E Sladewski; Aoife T Heaslip

ARTICLE TYPE: Research Article

Dear Dr Heaslip,

We have now reached a decision on the above manuscript.

As you will see, both reviewers are positive but raise a number of criticisms and points that prevent me from accepting the paper at this stage. They suggest, however, that a revised version might prove acceptable, if you can address their concerns. If you think that you can deal satisfactorily with the criticisms on revision, I would be pleased to see a revised manuscript.

Please upload both a 'clean' version of your Word file, along with a highlighted version clearly showing where you have made changes made in the revised manuscript. Please avoid using 'Tracked changes' in Word files as these are lost in PDF conversion.

I should be grateful if you would also provide a point-by-point response detailing how you have dealt with the points raised by the reviewers in the 'Response to Reviewers' box. Please attend to all of the reviewers' comments. If you do not agree with any of their criticisms or suggestions please explain clearly why this is so.

Reviewer 1

Advance summary and potential significance to field

This is a very nice and particularly well-written manuscript, which describes a comprehensive biophysical characterization of the apicomplexan parasite *Toxoplasma gondii* myosin F (MyoF). The work addresses a substantial gap in our understanding of myosins (in parasites but also in a broader context) and provides novel insights into how actin dynamics may be controlled in *T. gondii*. The authors combine recombinant expression and mostly simple but elegant biochemical/biophysical

assays with high-resolution parasite imaging to propose a model, in which the WD40-containing tail oligomerizes MyoF into a multi-headed complex capable of processive movement on TgAct1 and of sliding actin relative to microtubules.

The study is well executed, conceptually interesting, and will be of broad interest to readers working on molecular motors, cytoskeletal diversity, and apicomplexan parasite cell biology. However, I have some concerns and suggestions for improvement, detailed below.

Comments for the author

Major comments:

1. The evidence for conversion of a dimer into a ≥ 6 -headed oligomer via the WD40 domain is compelling but remains qualitative. Negative-stain EM and SEC support oligomerization, but the exact architecture (hexamer? trimer of dimers?) is still inferred. Given that this is central to the mechanistic model, I would have liked to see at least predictions (by AF) of the WD40 region in different oligomeric states. This would substantially strengthen this section, provide a useful structural framework for interpreting the EM classes, and guide future mutagenesis studies to test the proposed interfaces.
2. The observation that MyoF contains six IQ motifs but binds only three light chains is intriguing, and the demonstration that TgCaM alone vs. with TgMlc1 dramatically alters velocity is important. However, the mechanistic basis for differential light-chain occupancy and regulation remains speculative. Do you think that CaM might only bind to three IQs but the rest could still be capable of binding Mlc1 when required? Why wasn't the assay used to determine the heavy-chain-CaM ratio not used for Mlc1, too? Again, AF modelling of the IQ motifs with TgCaM and TgMlc1 (individually and in different combinations) could add some mechanistic insight. Such predictions could help reveal which IQ motifs are compatible with each light chain, whether Mlc1 induces conformational changes in the lever arm, or whether certain IQs are sterically incompatible with binding.
3. The link between TgAct1 D-loop structure and enhanced motility of MyoF on TgAct1 and jasplakinolide-stabilized bundles is compelling but somewhat indirect. It would be helpful to clarify whether the differences observed stem from intrinsic TgAct1 structural features or from stabilization by jasplakinolide. An additional control using unstabilized skeletal actin bundles would strengthen the argument.
4. The demonstration of both the full-length motor and the WD40 tail binding to microtubules and the observed intraconoid localization are intriguing. The functional implications remain somewhat speculative. Some modeling of the complex, clarifying ionic-strength dependence, and if possible, providing an estimated affinity or minimal binding region, would help define the physiological relevance of this interaction.
5. The model that a multi-headed MyoF complex slides or rearranges adjacent actin filaments is reasonable and supported by several lines of indirect evidence. Still, some elements remain inferred. It would help the reader if the authors explicitly distinguish between experimentally demonstrated behaviors (e.g., processivity on TgAct1, multi-headed oligomerization) and more speculative elements (e.g., sliding of native cytosolic actin networks in vivo).

Minor comments:

1. The definition of "tail" is a bit confusing. It seems like, here, it is meant to refer to the WD40 only. However, in myosins in general, tail refers to the entire coiled-coil (dimerization) and cargo-binding region. Also, what was the logic for choosing residues 1-1265 as the Δ WD40 construct? From the schematic presentation, it seems that that construct is cut in the middle of the "unknown" region and results in a deletion of a much larger chunk than just the WD40.
2. What structure(s) does AF predict for the "unknown" region between the rod and the WD40 domain?
3. What is meant by "...is used to functionalize..." on lines 95-96 (page 5)?

4. In the beginning of discussion: Shouldn't it be class 22 (or XXII), not 27? Also, there seems to be some mixed use of Roman and Arabic numerals for myosin classes throughout the manuscript.
5. In figure 1:
 - It would be nice to show CaM also in cartoon and maybe in a slightly better visible color.
 - For orientation purposes, it would be good to label some of the conserved parts in the motor domain and possibly try to find a better view or show two views 90° apart.
 - In the schematic figure, the last amino acid is labeled as 1953, but from the figure legend, one gets the impression the last residue is number 1952.
6. The figures involving actin stabilization (phalloidin vs. jasplakinolide) would benefit from clearer labeling of conditions.
7. Quantification of EM classes (e.g., distribution of observed head numbers) would improve clarity.
8. Please clarify the prevalence of pausing and "end retention" events in single-molecule runs.
9. Some minor typographical issues and repetitions should be corrected, for example, line 166 (page 7): "the frequency of movement and run length was dramatically enhanced" (should be "were") and other similar examples.
10. Finally, it would be nice to compare TgMyoF to other apicomplexan MyoFs (esp. the Plasmodium equivalent).

Conclusion:

This manuscript provides important new mechanistic insight into a divergent apicomplexan myosin and proposes a plausible model for how MyoF drives actin dynamics and organization in *T. gondii*. I believe that, with the proposed revisions, the manuscript will be well suited for publication in the Journal of Cell Science.

Reviewer 2

Advance summary and potential significance to field

Overall, this is a comprehensive and careful analysis of the MyoF motor - the range of in vitro experiments provides detailed information on how the motor could work. The finding that the tail can organize the motor into a mini-ensemble and also has a role MT anchoring that enables to myosin to slide actin filaments along MTs is quite exciting as it reveals new modes of myosin motor operation. These significant findings extend our understanding of how myosin motors can operate in vivo and will be of interest to many in the field.

Comments for the author

Toxoplasma (Tg) MyoF is a class 27 myosin important for vesicle transport, organelle positioning and apicoplast division. This myosin does not appear to be directly associated with vesicle or organelles but instead contributes to actin organization that supports vesicle motility. Here the authors present the first characterization of this evolutionarily divergent motor that has a long lever arm (6 IQ motifs), a region of coiled coil, and a WD40 domain at its C-terminus. MyoF can bind to two different light chains (LCs), TgCam (calmodulin) and MLC1 and the motor operates optimally with just 3 TgCam bound, indicating that MLC1 regulates (slows) the activity of this motor. Interestingly, the lever arm region (LA) appears to be shorter when only 3 TgCam are present. Full-length MyoF forms a small ensemble with six motors extending from a central WD40 core and the tail alone (coiled-coil + WD40) forms a hexamer. The full-length myosin moves optimally TgAct1 filaments where the D-loop was previously discovered to be in an open conformation (Hvorency et al, 2024, Nat Comm) and on jasplakinolide actin bundles. The tail region binds to microtubules (MTs) and when the full-length motor is bound to MTs it can slide skeletal actin filaments along them, albeit at a slow speed. Super resolution microscopy shows that MyoF-GFP motor forms punctae that are not associated with subpellicular MTs but some are in the center of the conoid where MTs are

present. Together, these results reveal that the motility of MyoF can differ based on its LC complement and the nature of its actin track (bundled vs single filament, open or closed D-loop conformation). It suggests a tempting model whereby it affects organelle motility and actin dynamics by sliding actin filaments along microtubules. How this might actually occur in vivo is not all that clear but the results provide another example of filament sliding by motors plays a role in intracellular motility.

The individual experiments are presented in a clear manner but it is difficult to synthesize and understand what all of the results mean given the way the paper is organized. For example, the LA is described as being 'shortened' before the EM data are presented. The bulk of the motility experiments are done with rabbit skeletal actin, understandably because Toxo actin is not so easy to work with. However, given that MyoF appears to favor movement on the D-loop open conformation of actin and also that it is not known if Toxo even has actin bundles, it becomes confusing to know what results are significant or physiologically relevant when there is a mix of actins used. There is a sense of 'back and forth' in the motor analysis that makes it hard to keep track of things. Improving the way the data are presented would help readers to follow things better. Even if the information might not be physiologically relevant (e.g. bundled actin) the analysis is a useful compilation of the overall MyoF motor properties. In summary, the paper would benefit greatly from reorganization to better focus on the main findings that link motor properties to physiological function.

Major comments [Please request additional experiments only if they are essential for supporting the conclusions; authors should be encouraged to highlight any claims that are preliminary or speculative, or to discuss any pitfalls or alternative interpretations in a 'Limitations' section]

-- New data are not necessarily required, what is mostly needed is some quantification and clarification in several places.

1) When the motor is co-expressed with only TgCam the neck region, composed of IQ motifs, the lever arm region appears strikingly shortened (Fig 4A). Has this shortening of the lever arm been observed for any other myosins? Are there anything notable sequence differences between the IQ motifs that might provide clues about how this happens? Have the authors tried to use Alpha Fold multimer to model LC binding with single or mixed LCs? This might not be informative but it could reveal something about the LC-IQ interactions that would explain the observed shortening/flexibility of the lever arm.

2) The authors Discuss that the number of IQ motifs does not always match with the number of LCs that bind to a lever arm (lines 374-380). They cite interesting work on *Leishmania Myo26* (Batters et al, 2012) and state that the third IQ of Myo10 does not bind a LC (citing Bird et al, 2014 - presumably the authors intended to say Myo15A). It should be noted that myosin LCs are not always CaM, as seen here, and some myosins even use the well-known myosin II LCs, ELC and RLC, as is the case for Myo15A and Myo7A (Bird et al citation; Hollo et al, 2023, JBC). Also, when LCs are unknown it is possible that standard efforts to identify them, such as IPs, may be done under conditions that do not favor binding of LCs.

3) The initial assays are performed with rabbit skeletal actin and then the authors point out that the D-loop of native TgAct1 is in the open conformation. They then carry out experiments in the presence of jasplakinolide that stabilizes the open D-loop conformation and a few select experiments with TgAct1. This makes one wonder why the majority of the motor assays, such as those in the initial part of the paper, are not done with jasplakinolide actin if that is really a good stand-in for the native actin.

For example, does the full-length motor move more frequently on the single TgAct1 filaments or is less frequent as seen with the normal or jasplakinolide bundles?

4) In several places the authors present 'single molecule' motility (e.g. Fig 5; lines 226 - 229) This is a bit confusing as presumably what is being characterized here is the MyoF trimer. Does this mean the native motor ensemble or a dimerized motor? If it is the former, is it appropriate to consider that a single molecule?

- 5) It is sometimes confusing when the authors discuss a shortened lever arm because it is not always clear if they mean that it is physically shorter (as seen in EM images) or that the region itself (typically IQ numbers) is shorter.
- 6) Is TgCam binding to MyoF Ca²⁺-sensitive? Does Ca²⁺ impact the motility of either the dimerized motor or full-length myosin associated with TgCam?
- 7) The first part of the Discussion briefly discusses WD40 domains in cytoskeletal proteins and then shifts into a summary of motors that slide filaments. The first part does not seem to lead the reader anywhere significant, given that WD40 domains are not particularly unusual and there does not appear to be any particular link between the TgMyoF and other cytoskeletal proteins with WD40 domains. WD40 domains typically form dimers so it would be useful to discuss what is known, if anything, about their role in self-association beyond dimer formation.
- 8) lines 175 - 185 The paragraph describes the interesting results from negative stain EM of the TgMyoF delta tail, Va chimera and the full length myosin (Fig 4). The TgMyoF lever arm is said to be approximately half the length of the Va lever arm and the globular tail as 10 - 12 nm in diameter. These values seem reasonable based on the obvious differences in the images shown, but they appear to be approximations and are not supported by any statistics or details on particle measurements.
- 9) line 190 The text states that particles of the full-length MyoF contain 'up to six' motor domains. Are hexameric particles always found or is there some variability? From the illustration and the images (Fig 4C) it appears that the stalk of the myosin is not dimerized - is that the case? Quantification would be helpful here.
- 10) lines 231 - 232 The observation that MyoF pauses at the end of an actin filament before it dissociates is interesting (as is the similar observation that actin stalls at the ends of filaments in the sliding assays) but what should the reader take from these observations? Is the pause time quite variable or it is typically around 5 sec as seen in Fig 5G. Is this something that occurs frequently so it might be physiologically relevant? Is it seen with both actin and jasp-actin?
- 11) Fig 6B The localization of MyoF in *T. gondii* is investigated using ExM and many 'discrete spots' are observed, some of which are associated with a fiber in the center of the conoid, but not with the subpellicular microtubules. Earlier images of live cells expressing a fluorescent fusion of MyoF showed a clear peripheral localization (Kellermeier & Heaslip, 2024). Are the results here consistent with earlier work? How big are the particles and is their size consistent with known cargo known to be moved by MyoF?
Is actin present in the inner conoid where MTs and MyoF are found?

Minor comments

It can be challenging to keep track of all of the findings, in part because of the way the data are presented (the order). The authors should consider providing a summary table (it could be in the Supplement) that has the results of the different motility assays presented in one place.

line 65 The initial description of MyoF says that there are 'three unique inserts' in the motor domain and then states that the second one is adjacent to the light chain found to the first IQ motif. The figure legend (pg 21) states that there are four inserts. It would be helpful to describe what region of the motor contains the other two or three inserts.

line 84 It would be helpful to indicate that the mEmerald fusion used for the pull-down is the full-length protein.

line 86 The text describes two low molecular weight bands in the mEmerald-MyoF IP eluate, one at 60 kD and the other at 15 kD. There is also a clear band above the 60 (55?) kD band. What are these higher molecular bands (perhaps one is tubulin?) and why does TgMlc1 appear to be absent?
line 99 Shouldn't it be the 'bac-to-bac' system, as in baculovirus?

line 108 The phrasing is a bit odd - it almost reads as if the TgCam and TgMlc1 are gliding in the assay.

line 124 The conclusion from the LC binding analysis is that MyoF 'binds half as many light chains' as myosin Va and fewer than the number of IQ domains. As written it sounds as if the MyoF only ever binds three LCs (Cam or Mlc1). This could be the case but the stoichiometry of bound LCs for the motor co-expressed with Cam and Mlc1 is not provided so the conclusion here should be qualified or, preferably, it would be useful to provide information about how many LCs are bound to the heavy chain that is co-expressed with both of them.

line 153 This first sentence of the paragraph refers to 'myosin motors with shortened lever arms' before this feature of TgMyoF is explained, and also states that these are 'optimized for movement on actin bundles'. References 29-31 are cited in support of this. Those papers show that Myo10 indeed moves optimally on actin bundles, however the lever (LA) arm is not shorter - while there are only 3 LC-binding IQ motifs, that region is followed by a rigid stable alpha-helix (SAH) is considered to be part of the LA and it extends the IQ region. This makes the overall LA region as long as that of Myo5. However, the difference in the orientation of the converter as seen in the structure (Ropars et al, 2016) results in a 'flatter' orientation of the LAs that favors moving on bundles.

line 160 Again, the authors make reference to the 'shortened lever arm' of TgMyoF without providing the reader with any discussion of this (which follows in the next section).

lines 162 - 167 The motility of teams of TgMyoF on actin bundles is described here, but unlike other assays no movies are presented. These are not essential but given the striking difference observed when compared to motility on single filaments or single motors it would be good to provide an example.

line 175 Please state whether the EM analysis is for MyoF co-purified with TgCam.

lines 236,237 and 248,249 The motors are described as being non-processive or not supporting processive movement. This means that they don't move on filaments at all or that only very short runs are observed?

line 262 Reference is made to Fig S5 - that figure appears to be missing.

line 267 The text states that tubulin consistently co-purifies with MyoF based on the presence of tubulin peptides in the mass spec analysis of several independent pull-downs from cell lysates (Table S1 and a band at 55 kD on gels of these samples, Fig S2E,F). Has the identity of the 55 kD bands observed in the samples been confirmed by blotting?

line 350 The change in the D-loop of actin seen in jasplakinolide-treated actin filaments is here referred to as a 'modification' which is not really the case (it suggests something like a PTM). It might be more appropriate to describe it as a conformation.

line 361 "MyoVc forms a parallel dimer through ITS..."

line 438 The affinity resin is presumably from Sigma, that should be indicated here.

line 451 Following TEV cleavage of His-Fascin the digest is said to be added to nickel select beads equilibrated in 0.5M imidazole. Is that correct? If so, it seems that the His-tag would not bind to the beads. Is the TEV itself also His-tagged or is it also in the final Fascin fraction?
What is the source of the TEV?

lines 498, 533 How were the motility data (gliding and single molecule) analyzed?

line 519 "... imaging rate of 0.05 to 0.2 seconds" - is this frame/time?

line 550 Movement OF actin along... (or Actin movement)

lines 698 - 699 The text refers to dotted lines in the depiction of the dimerized C-terminus of MyoF (coiled-coil region plus WD4) in Fig 4D, however those dotted lines are so faint that they are quite difficult to see.

line 699 What type of analytical size exclusion chromatography column was used (the information is not in the Methods)?

line 768 Delete this line that just has "3."

line 773 Ref 13 citation should be updated, it appears that work has now been published in mBio.

Fig S2A The TgMlc1 is labeled as being 24 kD yet it runs slower than that.

Movies (single particle tracking) It would be helpful to highlight a few of the moving Qdots with an arrow. It is not always easy to spot them at first glance.

First revision

Author response to reviewers' comments

Response to Reviewers:

We thank the reviewers for their thoughtful comments. We have responded to each point below. Reviewer comments are in blue italics, our response is in black.

Comments from the Reviewers:

Reviewer 1: SUMMARY OF THE ADVANCE MADE IN THIS PAPER AND ITS POTENTIAL SIGNIFICANCE TO THE FIELD

This is a very nice and particularly well-written manuscript, which describes a comprehensive biophysical characterization of the apicomplexan parasite *Toxoplasma gondii* myosin F (MyoF). The work addresses a substantial gap in our understanding of myosins (in parasites but also in a broader context) and provides novel insights into how actin dynamics may be controlled in *T. gondii*. The authors combine recombinant expression and mostly simple but elegant biochemical/biophysical assays with high-resolution parasite imaging to propose a model, in which the WD40-containing tail oligomerizes MyoF into a multi-headed complex capable of processive movement on TgAct1 and of sliding actin relative to microtubules.

The study is well executed, conceptually interesting, and will be of broad interest to readers working on molecular motors, cytoskeletal diversity, and apicomplexan parasite cell biology. However, I have some concerns and suggestions for improvement, detailed below.

SUGGESTIONS TO AUTHORS

Major comments:

1. The evidence for conversion of a dimer into a ≥ 6 -headed oligomer via the WD40 domain is compelling but remains qualitative. Negative-stain EM and SEC support oligomerization, but the exact architecture (hexamer? trimer of dimers?) is still inferred. ^{SEP} Given that this is central to the mechanistic model, I would have liked to see at least predictions (by AF) of the WD40 region in different oligomeric states. This would substantially strengthen this section, provide a useful structural framework for interpreting the EM classes, and guide future mutagenesis studies to test the proposed interfaces.

AlphaFold modeling of the MyoF WD40 domain (residues M1572-V1953) predicts a hexameric assembly organized as a trimer of dimers. The structure appears to be stabilized by extensive

hydrogen-bonding networks at conserved dimer interfaces, as well as secondary interfaces that link adjacent dimers into a higher-order hexamer. The predicted arrangement positions the N-termini of paired WD40 domains in close proximity, supporting a model in which the WD40 domain mediates stable homo-oligomerization of MyoF. We have included the AlphaFold model as well as insets showing the two predicted interfaces stabilized by hydrogen bonding in Figure S7 and Movie S9 and described these results on line 236...

“To gain insight into how the WD40 tail domain might facilitate oligomerization, we used AlphaFold to determine if the MyoF W40 domain could form a hexameric arrangement. The model predicts that pairs of WD40 domains form dimers stabilized by extensive hydrogen bonding networks (Fig. S7A-D, orange, Fig. S7E), positioning the N-terminal regions of each tail adjacent to one another and that these dimers can further organize into a trimer of dimers through additional inter-dimer hydrogen bonding, consistent with a hexameric assembly (Fig. S7A-D, green, Fig. S7F, Movie S9). Detailed analysis of the interfaces reveals that the same sets of amino acid residues mediate contacts across all three dimer interfaces, with an additional hydrogen bonding network stabilizing the trimeric assembly.”

2. The observation that MyoF contains six IQ motifs but binds only three light chains is intriguing, and the demonstration that TgCaM alone vs. with TgMlc1 dramatically alters velocity is important. However, the mechanistic basis for differential light-chain occupancy and regulation remains speculative. Do you think that CaM might only bind to three IQs but the rest could still be capable of binding Mlc1 when required? Why wasn't the assay used to determine the heavy chain-CaM ratio not used for Mlc1, too? Again, AF modelling of the IQ motifs with TgCaM and TgMlc1 (individually and in different combinations) could add some mechanistic insight. Such predictions could help reveal which IQ motifs are compatible with each light chain, whether Mlc1 induces conformational changes in the lever arm, or whether certain IQs are sterically incompatible with binding.

We agree with the reviewer that the light chain regulation is intriguing. Counting TgCam binding was possible because TgCam refolds after denaturation, while TgMlc1 remains predominately insoluble, thus precluding us from performing this experiment with TgMlc1 only. When this experiment is performed with both TgMlc1 and TgCam, TgMlc1 is again predominately in the pellet with all of TgCam in the supernatant. Interestingly, the amount of TgCam in the supernatant is decreased in the presence of TgMlc1 suggesting that TgMlc1 displaces one or more TgCam molecules from MyoF. This data is now included in the manuscript in Supplemental Figure 3G and described on line 132:

“When this experiment was performed with MyoF-Motor bound to TgMlc1 or TgCam and TgMlc1, TgMlc1 was predominately in the pellet precluding us from determining the number of TgMlc1 molecules that bind MyoF. However, when both TgMlc1 and TgCam are bound to MyoF, the amount of TgCam bound to MyoF decreases, suggesting that TgMlc1 displaces one or more TgCam molecules in order to bind MyoF (Fig. S3G).”

While it is possible that TgCam might bind 3 IQ motifs and TgMlc1 binds the remaining IQ's, we find this scenario unlikely due to the observation that TgCam alone supports motor function indicating that the lever arm is sufficiently stiffened for the motor to generate force.

To gain further insight into light chain binding, we performed AlphaFold-based structural modeling of TgCam and TgMlc1 binding to MyoF IQ motifs 1-3 to assess potential binding preferences. While both light chains are predicted to bind all three motifs, contact probability heatmaps and interface score analyses (see details in the legend of **Figure S4**) indicate preferential binding of TgCam to IQ1 and IQ3, and of TgMlc1 to IQ2. These differences correlate with predicted structural conformations and suggest that TgMlc1 may preferentially occupy IQ2 relative to TgCam. This analysis is described on line 138:

“To determine which IQ motifs are preferentially occupied by TgCam versus TgMlc1, we used AlphaFold to model the binding of each light chain to MyoF IQ motifs 1-3. The structural predictions indicated that both TgCam and TgMlc1 are capable of binding all three motifs (Fig. S4A). To assess potential binding preferences, we quantified light chain-IQ motif interactions by generating contact probability heatmaps and calculating interface scores based on minimum residue-residue distances (see Fig. S4 legend for details). These analyses revealed differential binding tendencies: TgCam is predicted to bind more strongly to IQ1 and IQ3, whereas TgMlc1 shows a higher predicted affinity for IQ2 (Fig. S4B).

Structurally, the reduced TgCam interaction with IQ2 appears to result from limited engagement of its N-terminal N-lobe, which adopts a more extended conformation in this complex (Fig. S4A). Conversely, TgMlc1 is predicted to interact with IQ2 with both lobes, leading to

conformational differences that likely influence binding strength. Similar open or extended light chain conformations have been reported in other myosin-light chain complexes (28).

Interface scores, calculated as the summed contact probabilities across the interaction surface further supported these predictions. Higher interface scores were observed for TgCam with IQ1 and IQ3, and for TgMlc1 with IQ2, consistent with the heatmap analysis (Fig S4B). Taken together, these analyses suggest that TgMlc1 may preferentially occupy IQ2, potentially limiting TgCam binding at this site.”

3. The link between TgAct1 D-loop structure and enhanced motility of MyoF on TgAct1 and jasplakinolide-stabilized bundles is compelling but somewhat indirect. It would be helpful to clarify whether the differences observed stem from intrinsic TgAct1 structural features or from stabilization by jasplakinolide. An additional control using unstabilized skeletal actin bundles would strengthen the argument.

To directly address whether jasplakinolide stabilization is required for or enhances MyoF processivity, we performed an additional single-molecule imaging experiment using unstabilized TgAct1 filaments labeled with the actin chromobody. Under these conditions, MyoF remained robustly processive, demonstrating that jasplakinolide does not appear to promote motility.

We argue from this that the enhanced processivity of MyoF on TgAct1 arises from intrinsic structural features of TgAct1 filaments, rather than its stabilization. We have clarified this point in the revised manuscript and include a movie (Movie S12) showing Full-length MyoF processivity on unstabilized TgAct1 filaments and discussed this in the results section on line 270:

“Structural studies have shown that actin-stabilizing agents trap actin in distinct conformational states (40). Our recent work demonstrated that, unlike skeletal actin, unstabilized TgAct1 filaments adopt an open D-loop conformation, and that jasplakinolide acts to further stabilize this open conformation (41) which may promote MyoF binding and processivity. Thus, to determine whether jasplakinolide itself was required for processivity, we imaged single MyoF complexes on unstabilized TgAct1 filaments labeled with the actin chromobody. MyoF remained processive under these conditions (Fig. S12), indicating that jasplakinolide does not artificially promote processivity and that the intrinsic structural features of TgAct1 support MyoF motility.”

4. The demonstration of both the full-length motor and the WD40 tail binding to microtubules and the observed intraconoid localization are intriguing. The functional implications remain somewhat speculative. Some modeling of the complex, clarifying ionic-strength dependence, and if possible, providing an estimated affinity or minimal binding region, would help define the physiological relevance of this interaction.

We previously demonstrated that both the MyoF-Tail and full-length MyoF bind statically to microtubules at 50 mM KCl, and that actin translocation along microtubules occurs at 80 mM KCl, indicating that these interactions remain stable under these ionic conditions. To test whether MyoF-microtubule binding persists at higher ionic strength, we performed actin-microtubule translocation assays at 150 mM KCl. Our results show that MyoF can still interact functionally with both actin and microtubules at this salt concentration, indicating that the MyoF-microtubule association is maintained at physiological ionic strength. These data are provided as Movie S20 and are discussed in the Results section on line 353:

“To determine whether the interaction between MyoF and microtubules persists at physiological ionic strength, we repeated the assay in the presence of 150 mM KCl. MyoF-mediated translocation was maintained under these conditions, indicating that MyoF is capable of associating with microtubules at physiological ionic strength (Movie S21).”

To gain insight on how the MyoF-Tail binds microtubules, we used AlphaFold to identify two candidate regions that may mediate microtubule interaction. The first one is a disordered segment in the “mid region” which is directly preceding the WD40 domain. This region (residues P1465-A1571) is predicted by AlphaFold to be unstructured as indicated by low pLDDT confidence scores. Interestingly, it is strongly basic (predicted pI = 11.38) and contains multiple R/K clusters, consistent with electrostatic microtubule-binding mechanisms described for other microtubule-associated proteins. The second putative microtubule interacting region is the WD40 hexamer. Electrostatic surface analysis of the hexameric WD40 assembly reveals a continuous electropositive surface spanning multiple β -propeller domains on two faces of the oligomer, which may bind electrostatically to microtubules. This data is presented in Fig. S8 and Fig. S12 and discussed in the discussion on line 427:

“Two regions of the MyoF tail may contribute to microtubule binding. First, the disordered region predicted by AlphaFold (Fig. S8; residues P1465-A1571), which exhibits low pLDDT confidence, contains multiple R/K pairs and is strongly basic (predicted pI = 11.38), consistent with electrostatic microtubule-binding mechanisms described for other microtubule-associated proteins

(46, 47). Alternatively, electrostatic surface analysis of the hexameric WD40 domain reveals a continuous electropositive surface spanning multiple β -propeller domains on two faces of the assembly (Fig. S12). This extensive basic surface could suggest a candidate microtubule-binding interface.”

5. The model that a multi-headed MyoF complex slides or rearranges adjacent actin filaments is reasonable and supported by several lines of indirect evidence. Still, some elements remain inferred. It would help the reader if the authors explicitly distinguish between experimentally demonstrated behaviors (e.g., processivity on TgAct1, multi-headed oligomerization) and more speculative elements (e.g., sliding of native cytosolic actin networks in vivo).

We thank the reviewer for this important suggestion. In the revised manuscript, we have reorganized and clarified the Discussion section entitled “MyoF forms an oligomer using its WD40 tail domain” to more clearly distinguish experimentally supported findings from speculative interpretations.

In the first three paragraphs (beginning on line 367), we now explicitly summarize the experimental data demonstrating MyoF oligomerization and processive motility on TgAct1 filaments. In the final paragraph of this section (beginning on line 404), we present the proposed model in which MyoF slides cytosolic actin filaments in vivo, and we have revised the text to explicitly frame this as a working model based on inference rather than direct experimental demonstration. The speculative nature of this proposed mechanism is now clearly stated to avoid overinterpretation of the data.

Minor comments:

1. The definition of “tail” is a bit confusing. It seems like, here, it is meant to refer to the WD40 only. However, in myosins in general, tail refers to the entire coiled-coil (dimerization) and cargo-binding region. Also, what was the logic for choosing residues 1-1265 as the deltaWD40 construct? From the schematic presentation, it seems that that construct is cut in the middle of the “unknown” region and results in a deletion of a much larger chunk than just the WD40.

We have now renamed the constructs as follows: MyoF 1-1265 = MyoF-Motor and the MyoF 987-1953 (CC+WD40) = MyoF-Tail. For clarity, the domains used for these constructs are now depicted in Fig. 1A. There were discrepancies in the predictions of the coiled-coil domain based on primary amino acid sequence compared with AlphaFold. AlphaFold predicts the coiled-coil region extends to amino acid 1251 so the deltaTail construct only extends 10 amino acids after the end of the coiled-coil. The domain organization schematic in Fig. 1 was based on the primary amino acid sequence while the coil-coil domain shown in Fig. 1E was based on AlphaFold. We have modified Fig. 1A to reflect the domain organization according to AlphaFold and updated the figure legend accordingly. Residues 1-1265 were chosen for the MyoF-Motor construct to preserve as much of the coiled-coil rod as possible in order to preserve the putative dimerization domain, ensuring that the minimal construct is two-headed.

2. What structure(s) does AF predict for the “unknown” region between the rod and the WD40 domain?

We examined the region between the predicted coiled-coil rod and the WD40 domain using AlphaFold multimer modeling. In this analysis, the coiled-coil region (residues E993-W1251) forms the expected α -helical bundle as a hexamer. The adjacent mid-region (residues I1252-A1571) is predicted to associate loosely with this helical bundle; however, confidence in this portion of the model is low.

Specifically, the segment spanning residues P1465-A1571 is predominantly scored in the low-confidence range (pLDDT <70, largely <50), and the overall pTM (0.27) and ipTM (0.29) scores indicate low confidence in the global arrangement. We show this model in Fig. S8 and discussed in on line 245:

“Given this predicted hexameric arrangement of the WD40 domains, we next investigated the positioning of the region between the coiled-coil and WD40 domains. AlphaFold predictions of the hexameric coiled-coil and mid region preceding the WD40 domain show that the coiled-coil forms an alpha-helical bundle (Fig. S8, grey), and residues 1252-1464 of the mid region form a series of short alpha-helices that associate with this bundle (Fig. S8, blue). However, the C-terminal portion of the mid region (residues P1465-A1571) is unstructured and predicted with low confidence, as indicated by pLDDT scores and low overall pTM and ipTM scores (Fig. S8, yellow). Together, these predictions support a model in which the coiled-coil region and the mid region positions the WD40 domains to form the circular density observed in negative-stain EM of the full-length MyoF, while the exact organization of the preceding mid region remains poorly defined.”

We thought this result had other additional interesting implications - first its potential role permitting dynamic rearrangements between extended and compact states (line 391), which might explain the observed heterogeneity in the number of heads in the full length motor by EM and second, its potential role in microtubule binding (line 429).

3. What is meant by "...is used to functionalize..." on lines 95-96 (page 5)?

By functionalize we meant attached a fluorophore or quantum dot. We have changed this to read "for fluorescent labeling" (line 103).

4. In the beginning of discussion: Shouldn't it be class 22 (or XXII), not 27? Also, there seems to be some mixed use of Roman and Arabic numerals for myosin classes throughout the manuscript.

MyoF was originally categorized as a class 22 myosin (Foth et al 2006 - 16505385) and was subsequently reclassified to class 27 in a more extensive phylogenetic analysis (Odrionitz and Kollmar 2007 - PMID: 17877792). This reference has been added to the discussion. We have chosen to use the more recent classification. We have changed myosin classes to Arabic numerals throughout the manuscript.

5. In figure 1:

*- It would be nice to show CaM also in cartoon and maybe in a slightly better visible color.
- For orientation purposes, it would be good to label some of the conserved parts in the motor domain and possibly try to find a better view or show two views 90° apart.*

A view of the motor domain rotated by 90° is now included in Fig. 1. CaM is now colored green to enhance contrast. All inserts are now labeled.

- In the schematic figure, the last amino acid is labeled as 1953, but from the figure legend, one gets the impression the last residue is number 1952.

The last amino acid is Val1953. This correction has been made in the legend of **Fig. 1**.

6. The figures involving actin stabilization (phalloidin vs. jasplakinolide) would benefit from clearer labeling of conditions.

Labels have been added to **Figures 2, 3 and 5** which clearly indicate the actin used for each experiment.

7. Quantification of EM classes (e.g., distribution of observed head numbers) would improve clarity.

The distribution of number of heads has been analyzed and plotted in a frequency histogram. This data is now shown in Fig. S6B and discussed on line 223:

"Quantification of visible heads revealed that most particles contained 3-6 heads (Fig. 4C, Fig. S6B). Because negative-stain EM often underestimates the number of observable heads due to particle orientation effects, these results may be consistent with a hexameric organization of the full-length MyoF complex."

8. Please clarify the prevalence of pausing and "end retention" events in single-molecule runs.

Yes, the data has been analyzed for frequency and length of pausing at the end of the filament. This is now presented as a cumulative frequency distribution in **Fig. S9** and presented in the results section on line 280.

"Analysis of motile events show that full-length MyoF reached the end of filaments and paused with a mean pause time of 1.5 ± 0.26 seconds (Fig. S9, blue)."

9. Some minor typographical issues and repetitions should be corrected, for example, line 166 (page 7): "the frequency of movement and run length was dramatically enhanced" (should be "were") and other similar examples.

Typos and repetitions have been corrected throughout the manuscript.

10. Finally, it would be nice to compare TgMyoF to other apicomplexan MyoFs (esp. the Plasmodium equivalent).

We have included a multisequence amino acid alignment of MyoF from multiple Apicomplexan parasites and a unicellular algae from the related eukaryotic supergroup Alveolata in the supplemental material (**Fig. S1**). Significant sequence differences are discussed briefly in the results section (line 82) and in more detail in the figure legend.

"An alignment between TgMyoF and MyoF motors from several related species indicate the domain organization is conserved, however motor domain insert 1 is only conserved in the most closely related species including Neospora and Eimeria, while inserts 2 and 3 are present in all species but vary in both length and sequence. Plasmodium species contain an additional insert at the junction between the motor domain and lever arm that is not found in other species (Fig. S1)."

Conclusion:

*This manuscript provides important new mechanistic insight into a divergent apicomplexan myosin and proposes a plausible model for how MyoF drives actin dynamics and organization in *T. gondii*. I believe that, with the proposed revisions, the manuscript will be well suited for publication in the Journal of Cell Science.*

Reviewer 2: SUMMARY OF THE ADVANCE MADE IN THIS PAPER AND ITS POTENTIAL SIGNIFICANCE TO THE FIELD

Overall, this is a comprehensive and careful analysis of the MyoF motor - the range of in vitro experiments provides detailed information on how the motor could work. The finding that the tail can organize the motor into a mini-ensemble and also has a role MT anchoring that enables to myosin to slide actin filaments along MTs is quite exciting as it reveals new modes of myosin motor operation. These significant findings extend our understanding of how myosin motors can operate in vivo and will be of interest to many in the field.

SUGGESTIONS TO AUTHORS

Toxoplasma (Tg) MyoF is a class 27 myosin important for vesicle transport, organelle positioning and apicoplast division. This myosin does not appear to be directly associated with vesicle or organelles but instead contributes to actin organization that supports vesicle motility. Here the authors present the first characterization of this evolutionarily divergent motor that has a long lever arm (6 IQ motifs), a region of coiled coil, and a WD40 domain at its C-terminus. MyoF can bind to two different light chains (LCs), TgCam (calmodulin) and MLC1 and the motor operates optimally with just 3 TgCam bound, indicating that MLC1 regulates (slows) the activity of this motor. Interestingly, the lever arm region (LA) appears to be shorter when only 3 TgCam are present. Full-length MyoF forms a small ensemble with six motors extending from a central WD40 core and the tail alone (coiled-coil + WD40) forms a hexamer. The full-length myosin moves optimally TgAct1 filaments where the D-loop was previously discovered to be in an open conformation (Hvorency et al, 2024, Nat Comm) and on jasplakinolide actin bundles. The tail region binds to microtubules (MTs) and when the full-length motor is bound to MTs it can slide skeletal actin filaments along them, albeit at a slow speed. Super resolution microscopy shows that MyoF-GFP motor forms punctae that are not associated with subpellicular MTs but some are in the center of the conoid where MTs are present. Together, these results reveal that the motility of MyoF can differ based on its LC complement and the nature of its actin track (bundled vs single filament, open or closed D-loop conformation). It suggests a tempting model whereby it affects organelle motility and actin dynamics by sliding actin filaments along microtubules. How this might actually occur in vivo is not all that clear but the results provide another example of filament sliding by motors plays a role in intracellular motility.

The individual experiments are presented in a clear manner but it is difficult to synthesize and understand what all of the results mean given the way the paper is organized. For example, the LA is described as being 'shortened' before the EM data are presented. The bulk of the motility experiments are done with rabbit skeletal actin, understandably because Toxo actin is not so easy to work with. However, given that MyoF appears to favor movement on the D-loop open conformation of actin and also that it is not known if Toxo even has actin bundles, it becomes confusing to know what results are significant or physiologically relevant when there is a mix of actins used. There is a sense of 'back and forth' in the motor analysis that makes it hard to keep track of things. Improving the way the data are presented would help readers to follow things better. Even if the information might not be physiologically relevant (e.g. bundled actin) the analysis is a useful compilation of the overall MyoF motor properties. In summary, the paper would benefit greatly from reorganization to better focus on the main findings that link motor properties to physiological function.

Major comments [Please request additional experiments only if they are essential for supporting the conclusions; authors should be encouraged to highlight any claims that are preliminary or speculative, or to discuss any pitfalls or alternative interpretations in a 'Limitations' section]

-- New data are not necessarily required, what is mostly needed is some quantification and clarification in several places.

1) When the motor is co-expressed with only TgCam the neck region, composed of IQ motifs, the lever arm region appears strikingly shortened (Fig 4A). Has this shortening of the lever arm been observed for any other myosins? Are there anything notable sequence differences between the IQ

motifs that might provide clues about how this happens? Have the authors tried to use Alpha Fold multimer to model LC binding with single or mixed LCs? This might not be informative but it could reveal something about the LC-IQ interactions that would explain the observed shortening/flexibility of the lever arm.

Sequence comparison of the IQ motifs reveals clear differences in charge distribution and residue composition. IQ1 and IQ3 contain clusters of basic residues flanking the canonical IQ consensus sequence, whereas IQ2 contains a distinct arrangement of polar and hydrophobic residues that deviate more substantially from the consensus (Fig. 1C). Thus, to explore whether these differences influence light chain preference, we modeled TgCam and TgMlc1 binding to each IQ motif using AlphaFold multimer. These models are shown in FigS4A. AlphaFold predicts that both TgCam and TgMlc1 are capable of binding all three motifs. To see if there were any differences in affinity we generated contact probability heatmaps and interface scores based on minimum residue-residue distances (see Fig. S4 legend for details). These analyses revealed differential binding tendencies: TgCam is predicted to bind more strongly to IQ1 and IQ3, whereas TgMlc1 shows a higher predicted affinity for IQ2. Structurally, the reduced TgCam interaction with IQ2 appears to arise from limited engagement of its N-lobe, which adopts a more extended conformation in this complex. In contrast, the binding of TgMlc1 looks more typical. This quantitation is shown in Fig. S4 and discussed on line 138:

“To determine which IQ motifs are preferentially occupied by TgCam versus TgMlc1, we used AlphaFold to model the binding of each light chain to MyoF IQ motifs 1-3. The structural predictions indicated that both TgCam and TgMlc1 are capable of binding all three motifs (Fig. S4A). To assess potential binding preferences, we quantified light chain-IQ motif interactions by generating contact probability heatmaps and calculating interface scores based on minimum residue-residue distances (see Fig. S4 legend for details). These analyses revealed differential binding tendencies: TgCam is predicted to bind more strongly to IQ1 and IQ3, whereas TgMlc1 shows a higher predicted affinity for IQ2 (Fig. S4B).

Structurally, the reduced TgCam interaction with IQ2 appears to result from limited engagement of its N-terminal N-lobe, which adopts a more extended conformation in this complex (Fig. S4A). Conversely, TgMlc1 is predicted to interact with IQ2 with both lobes, leading to conformational differences that likely influence binding strength. Similar open or extended light chain conformations have been reported in other myosin-light chain complexes (28).”

Lever arm length in myosins is generally determined by the number of occupied IQ motifs. Thus, in principle, other myosins that bind three light chains would be expected to have a lever arm of comparable length to MyoF. For example, MYO15-2 contains three light-chain binding domains and functions on actin bundles in stereocilia. In contrast, class V myosins bind six light chains, producing an extended lever arm that enables the motor to span the ~36 nm actin pseudo-repeat along a single filament. Myosin 10, which also binds three light chains and preferentially moves on actin bundles, has been shown by rotary shadowing EM to adopt a relatively compact lever arm conformation (Knight et al. PMID: 16030012), reminiscent of what we observe for MyoF. However, Myosin 10 can also assume an extended configuration through elongation of its single α -helical (SAH) domain. While it remains speculative, it is possible that motors specialized for movement on actin bundles, where lateral flexibility and access to adjacent filaments reduce the strict requirement for spanning a single filament repeat, may tolerate or even favor shorter or more flexible lever arms.

2) The authors Discuss that the number of IQ motifs does not always match with the number of LCs that bind to a lever arm (lines 374-380). They cite interesting work on Leishmania Myo26 (Batters et al, 2012) and state that the third IQ of Myo10 does not bind a LC (citing Bird et al, 2014 - presumably the authors intended to say Myo15A). It should be noted that myosin LCs are not always CaM, as seen here, and some myosins even use the well-known myosin II LCs, ELC and RLC, as is the case for Myo15A and Myo7A (Bird et al citation; Hollo et al, 2023, JBC). Also, when LCs are unknown it is possible that standard efforts to identify them, such as IPs, may be done under conditions that do not favor binding of LCs.

We have corrected the mistake of incorrectly referencing Myo10 instead of Myo15a. While we cannot exclude the possibility that other calmodulin-like proteins could bind MyoF in addition to Cam and MLC and we assume that in the presence of Cam the lever arm is fully occupied because it is sufficient to support motility. We have edited the discussion accordingly, starting on line 480.

3) The initial assays are performed with rabbit skeletal actin and then the authors point out that the D-loop of native TgAct1 is in the open conformation. They then carry out experiments

in the presence of jasplakinolide that stabilizes the open D-loop conformation and a few select experiments with TgAct1. This makes one wonder why the majority of the motor assays, such as those in the initial part of the paper, are not done with jasplakinolide actin if that is really a good stand-in for the native actin.

For example, does the full-length motor move more frequently on the single TgAct1 filaments or is less frequent as seen with the normal or jasplakinolide bundles?

While we predict that the D-loop conformation of jasplakinolide-stabilized skeletal actin may more closely resemble that of TgAct1 compared with phalloidin-stabilized filaments, several important considerations led us to use phalloidin-stabilized skeletal actin for the initial biophysical characterization.

First, fluorescent phalloidin-labeled skeletal actin is the field standard for single-molecule myosin assays, enabling direct comparison of MyoF behavior with previously characterized motors. Second, fluorescently labeled phalloidin allows reliable visualization of actin filaments, which is essential for accurate run length measurements. Although fluorescent jasplakinolide probes (e.g., SiR-actin) are commercially available, we find that these probes do not bind TgAct1 (unpublished observations). Moreover, we cannot exclude the possibility that fluorescently modified jasplakinolide alters filament structure differently than unlabeled Jas, making interpretation less straightforward.

To directly address the reviewer's concern regarding physiological relevance, we have now performed additional single-molecule assays using unstabilized TgAct1 filaments. Importantly, full-length MyoF remains robustly processive on native (unstabilized) TgAct1. These new data (Movie S12) demonstrate that jasplakinolide stabilization is not required for MyoF processivity and validate our conclusions regarding MyoF motility on TgAct1. This data is presented in the text on line 269: *“Structural studies have shown that actin-stabilizing agents trap actin in distinct conformational states (40). Our recent work demonstrated that, unlike skeletal actin, unstabilized TgAct1 filaments adopt an open D-loop conformation, and that jasplakinolide acts to further stabilize this open conformation (41) which may promote MyoF binding and processivity. Thus, to determine whether jasplakinolide itself was required for processivity, we imaged single MyoF complexes on unstabilized TgAct1 filaments labeled with the actin chromobody. MyoF remained processive under these conditions (Movie S12), indicating that jasplakinolide does not artificially promote processivity and that the intrinsic structural features of TgAct1 support MyoF motility.”*

With respect to bundle-based motility, our present study focuses on single-filament assays to define intrinsic motor properties. We agree that examining motility on higher-order filament architectures would be an interesting future direction but consider this beyond the scope of the current mechanistic analysis.

4) In several places the authors present 'single molecule' motility (e.g. Fig 5; lines 226 - 229) This is a bit confusing as presumably what is being characterized here is the MyoF trimer. Does this mean the native motor ensemble or a dimerized motor? If it is the former, is it appropriate to consider that a single molecule?

For the experiments with full-length MyoF, we use the term 'single molecule' when referring to a native motor ensemble. Although this term is a misnomer, we considered this appropriate as this is the standard in the field. There is a large body of literature where 'single molecule' references a single native motor ensemble. For example, 'single molecule' motility of MyoVa described in (PMID: 15764654), refers to a dimer of MyoVa heavy chain and 12 associated calmodulins. Dimeric kinesin, dimeric dynein or a complex of dynein, dynactin and BicD were all referred to a 'single molecule' by Ferro et al (PMID: 31498080), meaning a single native complex was attached to each fluorophore (e.g. Qdot or AlexaFluor). Having said that, we agree with the reviewers point that this might be confusing to some readers, thus we have reworded this sentence to read (line 257) *“Because full-length MyoF oligomerizes into a multi-headed complex, we first asked whether a single hexameric complex could move processively on actin filaments.”*. We have changed the phrase “single molecule” to “single complex” throughout this section of the manuscript.

5) It is sometimes confusing when the authors discuss a shortened lever arm because it is not always clear if they mean that it is physically shorter (as seen in EM images) or that the region itself (typically IQ numbers) is shorter.

We have edited the text to removed the references “to short/shortened lever arms”. We are more specific with our language and refer specifically to the number of IQ motifs in a given motor.

6) Is TgCam binding to MyoF Ca²⁺-sensitive? Does Ca²⁺ impact the motility of either the dimerized motor or full-length myosin associated with TgCam?

This is an incredible interesting question and one we are very interested in understanding. In *Toxoplasma*, Ca²⁺ signaling is a key regulator of processes such as parasite egress, which is accompanied by global rearrangements of the actin cytoskeleton. Given that light chains undergo conformational changes upon Ca²⁺ binding which affect motor function, Ca²⁺-dependent modulation of MyoF activity is certainly a plausible regulatory mechanism. In addition to calcium binding to light chains, other potential modes of regulation include heavy chain and light chain phosphorylation and function of the unique inserts. While interesting, addressing these questions is beyond the scope of this paper. We aim to address these interesting questions in a follow-up publication.

7) The first part of the Discussion briefly discusses WD40 domains in cytoskeletal proteins and then shifts into a summary of motors that slide filaments. The first part does not seem to lead the reader anywhere significant, given that WD40 domains are not particularly unusual and there does not appear to be any particular link between the TgMyoF and other cytoskeletal proteins with WD40 domains. WD40 domains typically form dimers so it would be useful to discuss what is known, if anything, about their role in self-association beyond dimer formation.

We have modified the discussion of WD40 domains starting on line 396. While many WD40 domains can mediate dimer formation, we only found a single example in the literature where WD40 domains can form trimers and find no examples of this domain mediating the formation of larger oligomers. We highlight an example of a WD40 domains forming trimers. From our searches of the literature, we have not found another example whereby this domain forms hexamers. This information is now included in the discussion paragraph.

8) lines 175 - 185 The paragraph describes the interesting results from negative stain EM of the TgMyoF delta tail, Va chimera and the full length myosin (Fig 4). The TgMyoF lever arm is said to be approximately half the length of the Va lever arm and the globular tail as 10 - 12 nm in diameter. These values seem reasonable based on the obvious differences in the images shown, but they appear to be approximations and are not supported by any statistics or details on particle measurements.

We have analyzed the negatively stained EM images and determined the average lever arm length for MyoF-5a and MyoF-Motor constructs. From these images, we find that the average lever arm length for MyoF-5a is 19.4 ± 0.6 nm, which is around what we would expect for a motor with 6 calmodulin-binding IQ motifs. We find that the average length for MyoF-Motor is 8.6 ± 0.5 nm which is approximately half the length of 5a. This data is consistent with our other data that estimates 3 bound light chains for MyoF. This data is shown in **Fig. S6A** and referenced on line 213.

9) line 190 The text states that particles of the full-length MyoF contain 'up to six' motor domains. Are hexameric particles always found or is there some variability? From the illustration and the images (Fig 4C) it appears that the stalk of the myosin is not dimerized - is that the case? Quantification would be helpful here.

Yes, we do see variability in the number of heads visible from these images. To quantify this, we counted the number of observable heads and plotted this data in a frequency distribution histogram. This data is now shown in **Fig. S6B** and discussed on line 224:

“Quantification of visible heads revealed that most particles contained 3-6 heads (Fig. 4C, Fig. S6B). Because negative-stain EM often underestimates the number of observable heads due to particle orientation effects, these results may be consistent with a hexameric organization of the full-length MyoF complex.”

Given the distribution, we modified our description of these particles from “up to 6” heads to “most particles contained 3-6 heads” which we think best describes the distribution of the data.

10) lines 231 - 232 The observation that MyoF pauses at the end of an actin filament before it dissociates is interesting (as is the similar observation that actin stalls at the ends of filaments in the sliding assays) but what should the reader take from these observations? Is the pause time quite variable or it is typically around 5 sec as seen in Fig 5G. Is this something that occurs frequently so it might be physiologically relevant? Is it seen with both actin and jasp-actin?

We feel that this is a very unique and interesting property for a myosin and why we wanted to include it in this paper. To further characterize this, we analyzed the length of pausing at the end of the filament. This is now presented as a cumulative frequency distribution in **Fig. S9** and presented in the results section on line 280:

“Analysis of motile events show that full-length MyoF reached the end of filaments and paused with a mean pause time of 1.5 ± 0.26 seconds (Fig. S9, blue).”

We also compared the pause time on skeletal actin bundles, which have a pause time of 4 ± 2.2 seconds. This data is also shown in **Fig. S9**. We also routinely observe end retention on rhodamine-labeled skeletal actin filaments for MyoF-Motor (**Movie S7**) so likely not due to Jas stabilization. While it is not yet clear what the function of end retention is in *T. gondii*, we have incorporated this property into our model (**Fig. 7**) as a possible way of sliding filaments relative to one another to facilitate actin dynamics within the cytosol.

11) Fig 6B The localization of MyoF in T gondii is investigated using ExM and many 'discrete spots' are observed, some of which are associated with a fiber in the center of the conoid, but not with the subpellicular microtubules. Earlier images of live cells expressing a fluorescent fusion of MyoF showed a clear peripheral localization (Kellermeier & Heaslip, 2024). Are the results here consistent with earlier work? How big are the particles and is their size consistent with known cargo known to be moved by MyoF?

The peripheral puncta of MyoF are ~200nm in size in the expanded image and therefore represent structures ~50nm or smaller prior to expansion. This is 5-times smaller than the size of dense granule vesicles that are moved in a MyoF dependent manner. There is the highest density of MyoF puncta in focal plane where the microtubules are in focus. This data is consistent with previous work where MyoF was found to have two localizations, a cytosolic pool and a peripheral pool where the motor was thought to be anchored into another cytoskeletal structure called the inner membrane complex (IMC). Therefore, we do not think these puncta represent MyoF molecules directly associated with vesicular cargo.

Is actin present in the inner conoid where MTs and MyoF are found?

Actin has not previously been observed inside the conoid. However, this could be due to technical limitations in imaging actin in *T. gondii*. First, the conoid is a small structure ~380nm in diameter and ~280nm in length. Therefore, it is exceedingly difficult to image structures inside the conoid using conventional microscopy. Imaging using higher resolution microscopy techniques has also not been feasible. *T. gondii* actin is highly dynamic and there is only a single cellular probe for imaging F-actin in cells (the Actin chromobody). Previous work from our lab (Kellermeier and Heaslip, 2024), showed that fixation disrupts actin organization which is usually required for high resolution imaging techniques. Further investigation of the roles of MyoF and actin at the intraconoid microtubule will be the focus of future studies.

Minor comments

It can be challenging to keep track of all of the findings, in part because of the way the data are presented (the order). The authors should consider providing a summary table (it could be in the Supplement) that has the results of the different motility assays presented in one place.

This table is now included or better organize the comparisons (**Table 2**)

line 65 The initial description of MyoF says that there are 'three unique inserts' in the motor domain and then states that the second one is adjacent to the light chain found to the first IQ motif. The figure legend (pg 21) states that there are four inserts.

The figure legend has been corrected to indicate 3 unique inserts.

It would be helpful to describe what region of the motor contains the other two or three inserts. We have included a sequence alignment where we have indicated the location of the three inserts (**Fig. S1**).

line 84 It would be helpful to indicate that the mEmerald fusion used for the pull-down is the full-length protein.

We now state that the endogenous gene was tagged with EmFP (line 92)

line 86 The text describes two low molecular weight bands in the mEmerald-MyoF IP eluate, one at 60 kD and the other at 15 kD. There is also a clear band above the 60 (55?) kD band. What are these higher molecular bands (perhaps one is tubulin?) and why does TgMlc1 appear to be absent?

We assume that one of the ~55kDa bands is tubulin, because tubulin consistently copurifies with MyoF in SF9 preps as determined by mass spec. We have not verified this with a second experimental method, such as western blot. It is unclear what the second band is as the top hits from the pulldown do not have a predicted size of 60kDa. TgMlc1 purified with MyoF in 2 of the 3 IP's that were performed but had a lower abundance of peptides and thus is likely below the limit of detection of silver-stained gel.

line 99 Shouldn't it be the 'bac-to-bac' system, as in baculovirus?

This error has been correct.

line 108 The phrasing is a bit odd - it almost reads as if the TgCam and TgMlc1 are gliding in the assay.

We have edited this sentence to read (line 114) "Using an in vitro motility assay, we found that complexes containing MyoF-Motor-TgCam-TgMlc1 moved filaments with an average speed of 0.24 $\mu\text{m}/\text{sec}$ "

line 124 The conclusion from the LC binding analysis is that MyoF 'binds half as many light chains' as myosin Va and fewer than the number of IQ domains. As written it sounds as if the MyoF only ever binds three LCs (Cam or Mlc1). This could be the case but the stoichiometry of bound LCs for the motor co-expressed with Cam and Mlc1 is not provided so the conclusion here should be qualified or, preferably, it would be useful to provide information about how many LCs are bound to the heavy chain that is co-expressed with both of them.

We now describe this experiment performed with MyoF bound with TgMlc1 or TgMlc1 and TgCam in combination (line 138). From this data and the AlphaFold modeling, we suggest that TgMlc1 displaces TgCam to bind MyoF. However, we have softened the language and indicate that future structural studies need to be performed to fully resolve this issue and determine the structure of the protein from amino acids 848 to 993 that encompasses the 6 predicted IQ motifs (line 491).

"Given that MyoF binds three TgCam molecules instead of six, as predicted by the number of identifiable IQ motifs, future structural studies are needed to resolve which IQ motifs are occupied by light chains and to determine the structure of the remaining unoccupied IQ motifs."

line 153 This first sentences of the paragraph refers to 'myosin motors with shortened lever arms' before this feature of TgMyoF is explained, and also states that these are 'optimized for movement on actin bundles'. References 29-31 are cited in support of this. Those papers show that Myo10 indeed moves optimally on actin bundles, however the lever (LA) arm is not shorter - while there are only 3 LC-binding IQ motifs, that region is followed by a rigid stable alpha-helix (SAH) is considered to be part of the LA and it extends the IQ region. This makes the overall LA region as long as that of Myo5. However, the difference in the orientation of the converter as seen in the structure (Ropars et al, 2016) results in a 'flatter' orientation of the LAs that favors moving on bundles.

We have edited this sentence to read (line 187) "Some classes of cargo transporting myosin motors have lever arms containing less than six IQs motifs and are optimized for movement on actin bundles rather than single actin filaments". The reviewers point is well taken that a shortened IQ motifs (as defined by the number of light chains bound) does not necessarily indicate that the lever arm itself is shorter and that the structure of this entire region could strongly influence track selection. We have added to the discussion to reflect this point. Line 494 now reads:

"As shown for Myo10, which contains 3 IQ motifs following by a single alpha-helix which are positioned in a manner to favor movement on bundles (30), structural determination of this region of MyoF will shed light on the orientation of the lever arm in relation to the motor domain would provide insight into MyoF's preference for bundled actin."

line 160 Again, the authors make reference to the 'shortened lever arm' of TgMyoF without providing the reader with any discussion of this (which follows in the next section).

We removed references of a shorted lever arm.

lines 162 - 167 The motility of teams of TgMyoF on actin bundles is described here, but unlike other assays no movies are presented. These are not essential but given the striking difference observed when compared to motility on single filaments or single motors it would be good to provide an example.

This video is now included in the manuscript (**Movie S14**)

line 175 Please state whether the EM analysis is for MyoF co-purified with TgCam.

This statement has been added to the Figure 4 legend and to the results on line 222.

lines 236,237 and 248,249 The motors are described as being non-processive or not supporting processive movement. This means that they don't move on filaments at all or that only very short runs are observed?

For the MyoF chimera, a small number of motors are observed transiently associating with actin, but no movement on actin filaments are observed (**Movie S17**). Since this experiment was performed using single molecule conditions (i.e. single molecules bound to a fluorophore) this data demonstrates that the MyoF/5a chimera is non-processive. The results have been rephrased for clarity on line 305:

“We also find that unlike the full-length motor, the MyoF/5a dimer is non-processive as a single motor on TgAct1 filaments (Movie S17). This result may indicate that the kinetic and thermodynamic properties of the MyoF motor domain likely requires oligomerization into a hexameric organization to support processivity.”

line 262 Reference is made to Fig S5 - that figure appears to be missing.

Our apologies for this oversight. Fig S11 is now included.

line 267 The text states that tubulin consistently co-purifies with MyoF based on the presence of tubulin peptides in the mass spec analysis of several independent pull-downs from cell lysates (Table S1 and a band at 55 kD on gels of these samples, Fig S2E,F). Has the identity of the 55 kD bands observed in the samples been confirmed by blotting?

The identity of these bands has not been confirmed by western blotting. Since tubulin was found routinely in the pulldowns and with Sf9 purification as identified my mass spec, we assume that tubulin band would recognize in a western blot of the IP since it is in the sample. However, this would not necessarily confirm that this is the identity of the most abundant band in the sample.

line 350 The change in the D-loop of actin seen in jasplakinolide-treated actin filaments is here referred to as a 'modification' which is not really the case (it suggests something like a PTM). It might be more appropriate to describe it as a conformation.

We now refer to this as confirmation rather than modification.

line 361 "MyoVc forms a parallel dimer through ITS..."

This typo has been corrected.

line 438 The affinity resin is presumably from Sigma, that should be indicated here.

The method now reads “Supernatant was added to 1 mL prepared nickel select affinity resin (Sigma Cat # P6611)....” (line 553)

line 451 Following TEV cleavage of His-Fascin the digest is said to be added to nickel select beads equilibrated in 0.5M imidazole. Is that correct? If so, it seems that the His-tag would not bind to the beads. Is the TEV itself also His-tagged or is it also in the final Fascin fraction?

As part of our protocol for HIS purification, we wash the HIS resin with 0.5 M Imidazole, then wash out the Imidazole before binding the protein. The Imidazole step allows us to use DTT during batch. I have made this clearer in the methods section on line 553.

“Supernatant was added to 1 mL prepared nickel select affinity resin (Sigma Cat # P6611) equilibrated with 0.5 M Imidazole followed by wash buffer to remove the Imidazole then and batch incubated at 4°C on a rocking platform for 45 minutes.”

What is the source of the TEV?

New England Biolabs (P8112S) - Source was added to the methods.

lines 498, 533 How were the motility data (gliding and single molecule) analyzed?

The analysis method was added in the Methods section on line 630. “Filament motility was analyzed using the FIJI plug-in MTrackJ plugin (67).”

line 519 “. imaging rate of 0.05 to 0.2 seconds” - is this frame/time?

This refers to the frame rate. This distinction has been fixed in the methods section on line 651 “imaging rate of 5 to 20 frames per seconds where indicated.”

line 550 Movement of actin along... (or Actin movement)

This typo has been corrected

lines 698 - 699 The text refers to dotted lines in the depiction of the dimerized C-terminus of MyoF (coiled-coil region plus WD4) in Fig 4D, however those dotted lines are so faint that they are quite difficult to see.

The line weight and color have been changed to make this visible.

line 699 What type of analytical size exclusion chromatography column was used (the information is not in the Methods)?

This information is now added to the methods section on line 758:

“Analytical size exclusion chromatography

MyoF-Tail and protein standards were dialyzed into gel filtration buffer (10 mM imidazole, pH 7.4, 300 mM NaCl, 1 mM EGTA, 1 µg/mL leupeptin, 1 mM DTT). Samples (400 µL) containing either 0.25 mg/mL MyoF-Tail or 0.5 mg/mL protein standards (see Fig. 4E legend) were clarified by centrifugation at 30,000 × g for 20 min at 4°C and loaded separately onto a Superdex 10/300 GL size-exclusion column equilibrated in the same buffer.”

line 768 Delete this line that just has "3."

This error has been corrected

line 773 Ref 13 citation should be updated, it appears that work has now been published in mBio.

This reference has been updated

Fig S2A The TgMlc1 is labeled as being 24 kD yet it runs slower than that.

The predicted size of TgMlc1 is 24kDa, yet runs at an apparent size of 30kDa. This is now indicated in the figure legend.

Movies (single particle tracking) It would be helpful to highlight a few of the moving Qdots with an arrow. It is not always easy to spot them at first glance.

Several movies showing negative results—for example, those in which single molecules of TgMyoF exhibit no motion—are included for completeness. To aid clarity, we have indicated in the corresponding movie legends which recordings show no observable events.

Second decision letter

MS ID#: jcs.264520R1

MS Title: Molecular features of Myosin F adapted for driving actin flows in *Toxoplasma gondii*

Authors: Thomas E Sladewski; Aoife T Heaslip

Article Type: Research Article

Dear Aoife,

I feel there is no need to go back to the reviewers, after your extensive revision in which you addressed all the reviewers concerns, so I am happy to tell you that your manuscript has been accepted for publication in Journal of Cell Science, pending standard publication integrity checks.